# A Driving-Style-Adaptive Framework for Vehicle Trajectory Prediction

**Di Wen**[1,2,3]    **Yu Wang**[2]    **Zhigang Wu**[1,3]
**Zhaocheng He**[1,2,3*]    **Zhe Wu**[2,*]    **Zheng Qingfang**[2]
[1]Sun Yat-sen University    [2]Pengcheng Laboratory
[3]Guangdong Provincial Key Laboratory of Intelligent Transportation System
{wend25,wuzhig6}@mail2.sysu.edu.cn   hezhch@mail.sysu.edu.cn
{wangy12,wuzh02,zhengqf01}@pcl.ac.cn

## Abstract

Vehicle trajectory prediction serves as a critical enabler for autonomous navigation and intelligent transportation systems. While existing approaches predominantly focus on pattern extraction and vehicle-environment interaction modeling, they exhibit a fundamental limitation in addressing trajectory heterogeneity originating from human driving styles. This oversight constrains prediction reliability in complex real-world scenarios. To bridge this gap, we propose the Driving-Style-Adaptive (**DSA**) framework, which establishes the first systematic integration of heterogeneous driving behaviors into trajectory prediction models. Specifically, our framework employs a set of basis functions tailored to each driving style to approximate the trajectory patterns. By dynamically combining and adaptively adjusting the degree of these basis functions, DSA not only enhances prediction accuracy but also provides **explanations** insights into the prediction process. Extensive experiments on public real-world datasets demonstrate that the DSA framework outperforms state-of-the-art methods.

## 1 Introduction

Vehicle Trajectory Prediction (VTP) serves as a fundamental capability for numerous intelligent transportation applications, including autonomous driving systems [1, 2, 3], motion planning algorithms [4, 5] and adaptive traffic control frameworks [6, 7]. Recent advances in VTP achieve notable progress through two primary paradigms: (1) capturing temporal patterns from historical trajectories and modeling vehicle interactions [8, 9, 10, 11], and (2) leveraging structured scene representations that incorporate road topology and regulatory constraints [12, 13, 14, 15]. However, these methods often overlook the originator of the trajectory: human drivers [16, 17], whose diverse behavior leads to heterogeneous trajectory patterns.

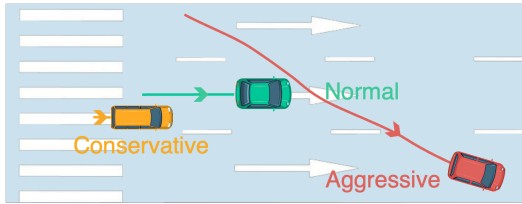

Figure 1: Illustration of three driving styles: Conservative drivers typically move slowly or stop to avoid obstacles; Aggressive drivers often travel at high speeds and are prone to overtaking other vehicles; Normal drivers maintain a constant speed and frequently change lanes to ensure safety.

In this paper, we propose an **adaptive** VTP framework based on distinct **driving styles** [18, 19]: **C**onservative, **A**ggressive and **N**ormal (CAN). Each driving style manifests in characteristic trajectory

---

*Joint Corresponding Authors.

39th Conference on Neural Information Processing Systems (NeurIPS 2025).

patterns, as illustrated in Figure 1. limited variability (conservative), non-smooth trajectories (aggressive) and frequent yet smooth motion changes (normal). For each styles, we employs variable basis functions within Kolmogorov-Arnold Networks (KANs) [20] to capture these trajectory patterns. In complex real-world scenarios, driver behavior often reflects a probabilistic mixture of weighted driving styles [21].

Our framework comprises two core components: (1) the matching between driving styles and their corresponding basis functions, and (2) the weighted combination and adjustment of the degrees of these functions. Additionally, inspired by the Weierstrass Approximation Theorem [22], our proposed DSA framework extends KANs from a theoretical perspective. Each matching in (1) is further grounded in the mechanical properties of basis functions, thereby providing explanations for our DSA framework.

Our main contributions in this paper are summarized as follows:

- To address the vehicle trajectory prediction task, we propose for the first time, a novel Driving-Style-Adaptive (DSA) framework tailored to the **driving styles** of human drivers and effectively leverages trajectory information.

- We utilize polynomial **approximation operators** to approximate and predict trajectories under different driving styles: Conservative, Aggressive and Normal (CAN). These operators support a **mathematical explanation** matching mechanism that matches each driving style with a corresponding polynomial form.

- The experimental results on real-world datasets (nuScenes, Argoverse and Waymo) demonstrate that our model significantly outperforms existing methods in vehicle trajectory prediction.

## 2 Preliminary & Related Work

### 2.1 Task Definition: Vehicle Trajectory Prediction (VTP)

VTP aims to predict the future trajectory of vehicles based on history trajectory or other informations available in a given scenario. In recent years, deep learning based VTP methods are categorized into two groups [23]: (i) knowledge-based methods, which incorporate specific information such as maps [24, 25], vehicles [9, 26] and interactions [27, 28] to represent the environment or vehicle behaviour. (ii) knowledge-free methods, which rely on deep learning's ability to encode complex data features, modeling them using structures such as tensors [29, 30] or attention mechanisms [31, 32].

Following above works, we analyze traffic scenes involving $N$ vehicles (agents). The trajectory of each vehicle $i$ in historical interval $[0, T]$ is denoted as $X_i = \left\{ s_i^{-t}, \cdots, s_i^0 \right\}$. Each state $s_i^\star$ is a 5-dimensional vector representing the $(x, y)$ position, velocity, acceleration, and the nearest lane segments ID. The superscript $\star$ denotes the time step. Similarly, the future trajectory in interval $[0, T]$ is given by $Y_i = \left\{ s_i^0, \cdots, s_i^T \right\}$.

### 2.2 Basic Network: Kolmogorov-Arnold Networks (KANs)

KANs [20] are inspired by the mathematical principles [33, 34, 35] of the Kolmogorov-Arnold representation theorem [36, 37, 38], stated as follows:

**Theorem 2.1** (**Kolmogorov-Arnold representation Theorem**) *For any multivariate continuous function* $f : [0, 1]^n \to \mathbb{R}$, $f$ *can be represented as a finite composition of univariate continuous functions* $\phi_{ij} : [0, 1] \to \mathbb{R}$ *and* $\Phi_j : \mathbb{R} \to \mathbb{R}$, *with the binary operation of addition such that:*

$$F = f(x_1, \cdots, x_n) = \sum_{j=1}^{2n+1} \Phi_j \left[ \sum_{i=1}^{n} \phi_{ij} (x_i) \right]. \tag{1}$$

The key innovation of KANs lies in implementing the residual activation functions $\phi(x)$ in Equation (1) as:

$$\phi(x) = w_b b(x) + w_s \psi(x), \tag{2}$$

where $b(x) = \mathrm{silu}(x)$ and $\psi(x) = \mathrm{spline}(x)$. Unlike Multi-Layer Perceptrons (MLPs [39, 40]), which utilize fixed activation functions associated with nodes ("neurons"), KANs feature learnable

$\phi(x)$ on edges ("weights"). However, due to the inherent complexity of these functions, the speed and scalability of the original KANs are not satisfactory [41]. Consequently, a variety of KAN-based applications are emerged in AI4Science tasks [42, 43, 44, 45, 46, 47]. To the best of our knowledge, we are the first to extension KANs to the VTP task. We achieve this by expanding the set of basis functions $\psi(x)$ to match different driving styles and by grounding this matching in both mathematical theory and task-specific behavior.

## 2.3 Core Theory: Applying Polynomials as Basis Functions

As describe in Section 2.2, a fixed basis function has inherent limitations. In the VTP task, such a function may fail to adequately approximate diverse trajectory or lane curves. This raises an important question: how can we legitimately expand the class of basis functions? In approximation theory, a fundamental question is whether polynomials can approximate any given continuous function to an arbitrary degree of precision. Weierstrass [22] provides a definitive answer by:

**Theorem 2.2** (*Weierstrass Approximation Theorem*) *Let* $f(x) \in L^p[0, 1]$ *for any* $p > 0$. *Then there exists an algebraic polynomial* $p_n(x) = \sum_{m=0}^{n} c_m x^m$ *such that*

$$\lim_{n \to \infty} \int_0^1 |f(x) - p_n(x)|^p \, dx = 0. \tag{3}$$

This interval can be extended to $[a, b]$. This demonstrates that polynomials $p_n \in \mathcal{P}_n$ can serve as basis functions $\psi(x)$ in Equation (2) to approximate the $f$ in Theorem 2.1. In our task, the vehicle trajectories are treated as the function $f$, with respect to the time step $t$. For different driving styles, we employ corresponding $p_n$ as basis functions to approximate these trajectories in accordance with Theorem 2.2. Furthermore, these trajectories also belong to the $L^p$ space[2], and are thus well-defined.

# 3 Methodology

## 3.1 Motivation and Overview

Our **D**riving-**S**tyle-**A**daptive (DSA) framework (illustrate in 2) models the behavior of the trajectory originator: the human driver. The **driving style** of various vehicle drivers are categoried as: **C**onservative, **A**ggressive and **N**ormal (CAN) [48, 49, 50], each reflecting distinct trajectory characteristics.

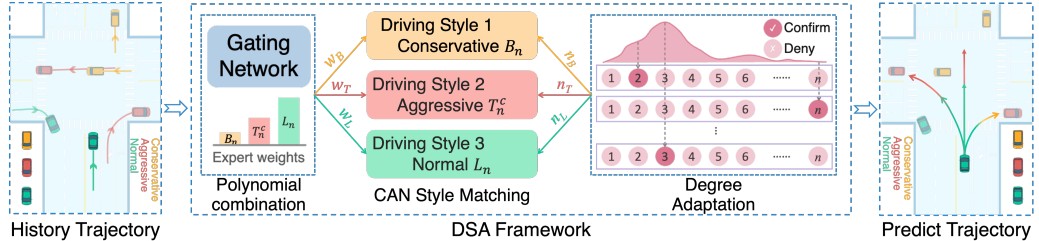

Figure 2: An overview of our DSA framework, which performs trajectory prediction based on driving style categories: conservative, aggressive and normal (CAN) to prediction. For clarity, we illustrate this process using a single vehicle example. the solid line represents trajectories length while the arrow ndicates the direction (history or future). The symbol $B_n$, $T_n^c$ and $L_n$ denote different basis functions $p_n$ corresponding to each driving style. Our proposed DSA framework dynamic adaps the experts (driving style) weighs $w_*$ and the degree $n_*$ of selected $p_n$.

We match each driving style characteristic to a corresponding approximation polynomial $p_n$ based on the mathematics properties of $p_n$ as described in Section 3.2. To implement this mechanism, we introduce $p_n$ combination and degree adjustment strategies in Section 3.3.

---

[2]Taking the $x$-position as an example, it can be shown that the integral $\int_\Omega |x(t)| \, dt < \infty$ holds

## 3.2 Theoretical Foundations for Matching Polynomials to Driving Styles

In this section, we elucidate the matching between the polynomials $p_n$ ($p_n \in \mathcal{P}_n$) and driving style, focusing on the mathematical properties of $p_n$ and analyzing the characteristics of each driver's type. Specifically, we address conservative drivers in Section 3.2.1, aggressive drivers in Section 3.2.2 and normal drivers in Section 3.2.3.

### 3.2.1 Conservative Drivers

Conservative drivers [51] prioritize driving comfort and safety, which leads to more cautious decisions. Their average speed is typically the slowest and rarely change their behavior. Consequently, their trajectories are characterized by smoothness and stability, with minimal abrupt changes in speed.

In this situation, we require a $p_n$ to capture approximating drivers with minimal behavioral changes, that is, ensures the approximation error decreases uniformly across the entire interval. we employ the Bernstein operatore[3] $B_n$ [52] to achieve this:

**Definition 3.1** (**Bernstein polynomial,** $B_n$) *Consider a function $f(x) \in C[0,1]$, $x \in [0,1]$, $B_n$ is specified by the equation:*

$$(B_n) f(x) = \sum_{k=0}^{n} f\left(\frac{k}{n}\right) \binom{n}{k} x^k (1-x)^{n-k}.$$

It is clear that $B_n \subseteq \mathcal{P}_n$ thus applies to Theorem 2.2. The primary advantage of the $B_n$ is articulated in the following proposition:

**Proposition 3.2** *For all functions $f \in C[0,1]$, the sequence $\{B_n f; n = 1, 2, 3, \cdots\}$ converges uniform[4] to $f$ as $B_n(f) \rightrightarrows f(x)$.*

This proposition demonstrates that the $B_n$ exhibits uniform convergence across the entire interval, making it particularly suitable for approximating trajectories with slow travel speeds and few behavioral changes, such as those of conservative drivers. This ensures that the approximation error decreases uniformly throughout the interval.

### 3.2.2 Aggressive Drivers

Aggressive drivers [53] prioritize their own benefits at the expense of safety and comfort, which leads to higher speeds, abrupt changes in acceleration and braking, with a frequent tendency to change lanes. As a result, their trajectories display more abrupt motions and are less smooth.

Trigonometric polynomials $T_n$ are dense[5] in $C(I)$ on the unit circle according to the Stone-Weierstrass Theorem [54]. This implies that trigonometric polynomials $T_n$ are particularly effective at approximating functions with discontinuities or sharp features, which define as: $T_n(x) = a_0 + \sum_{n=1}^{N} [a_n \cos(nx) + b_n \sin(nx)]$. We employ the Chebyshev polynomials [55] $T_n^c$, defined as follows:

**Definition 3.3** (**Chebyshev Polynomials,** $T_n^c$) *For $x \in [-1,1]$, the n-th $T_n^c$ of the first kind is given by $T_n^c(x) = \cos[n \cdot arccos(x)]$.*

The effectiveness of $T_n^c$ is further highlighted by the Chebyshev Minimax Theorem [56]:

**Theorem 3.4** (**Chebyshev Minimax Theorem**) *For $f \in C[-1,1]$, $T_n^c$ minimizes the maximum error in the uniform norm compared to any other $p_n$ approximation of the same degree. Formally, this relationship is expressed as:*

$$\|f - T_n^c\|_{L^\infty} \leqslant \|f - p_n\|_{L^\infty}.$$

---

[3]In this task, we leverage $p_n$ approximation operators to approximate and predict the vehicle trajectories, Specifically, we instantiate these operators using representative basis such as $B_n$, the same as other two approximation operators.

[4]**Uniform Convergence** For every $\epsilon > 0$, there exists an $N \in \mathbb{Z}^+$, $N = N(\epsilon)$, s.t. for all $n \geqslant N$, there is $|f_n(x) - f(x)| < \epsilon$.

[5]**Dense** A subset $A$ of a topological space $X$ is said to be dense in $X$ if every point of $X$ either belongs to $A$ or else is arbitrarily "close" to a member of $A$.

Theorem 3.4 explicitly states that $T_n^c$ can minimize the maximum error, effectively reducing the impact of sudden behavioral and speed changes typical of aggressive drivers. Furthermore, the overall prediction error is decreased.

### 3.2.3 Normal Drivers

Normal drivers [57] strike a balance between conservative and aggressive driving styles, representing a relatively common group in driving behavior. Their speed and acceleration typically fall between those of conservative and aggressive drivers, exhibiting moderate speed changes and occasional rapid reactions. Consequently, their trajectories are neither as smooth as those of conservative drivers nor as abrupt as those of aggressive drivers, but their trajectories may exhibit regular fluctuations.

This driving characteristic is closely related to the application of orthogonal polynomials $p_n^o$ [58]. The $p_n^o$ has significant flexibility and enables accurately capture the trajectories characterized by gradual changes and moderate fluctuations for normal drivers. The $p_n^o$ with weight function[6] $\rho$ and $\partial\left(p_n^o\right) = n$ are defined as: $\left\langle p_i^o, p_j^o \right\rangle = \int_a^b \rho\left(x\right) p_i^o\left(x\right) p_j^o\left(x\right) \ dx = \delta_{ij}$, where $\delta_{ij}$ equal to 0 *iff* $i \neq j$. The approximation of $p_n^o$ can be effectively described by [59]:

**Theorem 3.5** *(**Least Squares Characterization Theorem**) For any function $f \in C[a, b]$, there exists an orthogonal polynomial $p_n^o$ with $\partial\left(p_n^o\right) \leqslant n$ that minimizes the error in the $L_\rho^2$ norm[7] between $f(x)$ and $p_n^o(x)$:*

$$\|f - p_n^o\|_{L_\rho^2} = \min_{p \in \mathbb{P}_n} \|f - p\|_{L_\rho^2},$$

*where $\mathcal{P}_n$ denotes the space of all polynomials of degree at most $n$.*

The term "least" here does not denote the non-uniqueness, but rather indicates the possible to select optimal coefficients under best $L_\rho^2$ approximating. For instance, Legendre polynomials $L_n$ is a typical orthogonal polynomial:

**Definition 3.6** *(**Legendre Polynomial,** $L_n$) For $x \in [-1, 1]$ with a constant weight function $\rho\left(x\right) = 1$, $L_n$ is defined by*

$$L_n = \frac{1}{2^n n!} \frac{d^n}{dx^n} \left[\left(x^2 - 1\right)^n\right].$$

This orthogonality under the $L^2$ norm particularly without weight or a constant weight, makes it an exceptionally efficient tool for approximation [60], which represents a specific instance covered by Theorem 3.5. Moreover, $L_n$ is also defined by a simple recurrence relation:

$$(n + 1)L_{n+1}\left(x\right) = (2n + 1)xL_n\left(x\right) - nL_{n-1}\left(x\right),$$

This recurrence relation facilitates quick calculations and the optimal square approximation property excels under the $L^2$ norm. These characteristics make it well-suited for handling smooth and continuous trajectory fluctuations, align well with the The normal drivers. Their characterized by gradual changes and smoother transitions.

## 3.3 Algorithm Realization

### 3.3.1 Polynomial Combination

In Section 3.2, we utilize different polynomial forms to match different driving styles, thereby fully leveraging trajectory information for prediction. However, assuming a single fixed driving style may be inadequate in complex real-world scenarios. Kernel density estimation and latent variable analysis, reveal that driver behavior varies continuously with context and can be characterized as a probabilistic mixture [21, 61, 62] of weighted driving styles. Here we employ a MoE-TopK [63] approach to model multiple driving styles for trajectory prediction.

---

[6]**Weight Function** In open interval $(a, b)$, the defined positive, continuous, and integrable function is called weight function.

[7] **The $L_\rho^2$ Norm** Used to measure the "magnitude" or "error" of a function when combined with a particular weighting function $\rho(x)$ in a given interval. It is defined as $\|f\|_{L_\rho^2} = \left(\int_a^b |f(x)|^2 \rho(x) \, dx\right)^{1/2}$.

The process of combining the polynomials corresponding to multiple driving styles is presented in the algorithm on the right.. Here $X_i$ represents the $i-th$ history trajectory in $N$ vehicles as described in Section 2.1, which has 5 dimensions as $(x, y)$ position, velocity, acceleration, and the nearest lane segments ID. Experts represents the polynomials in Section 3.2. The output $z^{\text{Com}}$ is the feature of combine. In line 2, "SN" and "Sp" denote the Standard Normal and Softplus functions [64, 65], respectively, $W_g$ and $W_n$ are trainable weight matrices. In line 3, we define $H = (H_1, H_2, H_3)$.

This combination structure of $p_n$ allows each $E_i$ to better extract the trajectory feature in different driving styles, and enables the use of various basis functions for predict vehicle trajectory. To encourage all experts to contribute the combination process, Shazeer N et al. [63] introduce a load balancing loss function $L_{\text{MoE-K}}$ to encourage experts have equal importance as: $L_{\text{MoE-K}} = w_{\text{load}} \cdot$ CV $(\text{loads})^2$, where "CV" denotes the coefficient of variation.

---

**Algorithm: Polynomial Combination**

**Require:** Input vehicle trajectory $X_i$,
  1: Experts networ $\{E_j\}_{j=1}^3$, Gating network $G$
**Ensure:** Feature $z_i^{\text{Com}}$ via Polynomial Combination
  2: $H_j \leftarrow (X \cdot W_g)_i + \text{SN}() \cdot \text{Sp}[(X \cdot W_n)_i]$ for all $i$
  3: $G_j(x) = \text{Softmax}(H)$
  4: **for** $i = 1$ to $N$ **do**
  5:     $z_i^{\text{Com}} \leftarrow G_j(X_i) \cdot E_j(X_i)$.
  6: **end for**
  7: $z^{\text{Com}} \leftarrow \sum_{i=1}^N z_i^{\text{Com}}$

---

### 3.3.2 Degree Adjustment

Different driving style of trajectories can be approximated by corresponding $p_n$. However, the fixed degree of $p_n$ can restrict their ability for prediction entire trajectory of vehicles, which refers to:

**Theorem 3.7** (***Kolmogorov Theorem***) *For $f \in C[a, b]$, there exists a polynomial $p_n$ such that approximation error is bounded by:*

$$\|f - P_n\|_{L^\infty} \lesssim \left(\frac{\log n}{n}\right) V(f, [a, b]),$$

*where $V(f, [a, b])$ denotes the total variation[8] of $f$ over the interval.*

From Theorem 3.7, the accuracy of the polynomial approximation is directly related to the degree $n$ of the $p_n$, which applies broadly to $L^p$-space. On the other hand, the $n$ is bounded when error bounded of $p_n$ is know, this assertion is proved in Appendix C.

Adapting $n$ presents a complex non-convex and combinatorial optimization problem. To tackle this issue, we utilize SMAC3 [66] tool, which is particularly suitable for optimizing low-dimensional and continuous functions, suitable for characteristic of vehicle trajectory (Section 2.1). Specifically, the degree $n$ is treated as a hyperparameter optimization problem, aimed at minimizing the loss ($L$) on validation data $D_{\text{val}}$ and training data $D_{\text{train}}$. This process can be formulated as follows:

$$n_{\text{SMAC}} \in \arg\min_{n \in \mathbb{Z}^+} c(n) = \arg\min_{n \in \mathbb{Z}^+} L(\mathcal{D}_{\text{train}}, \mathcal{D}_{\text{val}}; n),$$

The hyperparameter optimization process targets the final degree $n_{\text{SMAC}}$, corresponding to achieve the least error for the corresponding basis function $p_n$.

## 4 Experiments

### 4.1 Basic Setting

We evaluate our DSA framework on three real-world vehicle trajectory prediction datasets: nuScenes [67], Argoverse [68] and Waymo [69]. These timestep settings follow the format(history time $\rightarrow$ prediction time): $2 \rightarrow 6$, $2 \rightarrow 3$ and $1 \rightarrow 8$, respectively. We utilize $L_{\text{oss}} = \lambda_1 L_{\text{Dis}} + \lambda_2 L_{\text{MoE-K}}$, with $L_{\text{MoE-K}} = w_{\text{load}} \cdot \text{CV}(\text{loads})^2$ for model training with balanced weighting parameters $\lambda_*$. We employ common standard metrics as the Average / Final Displacement Error (ADE / FDE) for evaluate generate $k$ trajectories. More detail of datasets and metrics, please refer to Appendix B.

---

[8]**Total Variation** A measure of the total amount of variation in a function over a given interval $[a, b]$, which is defined by $\sup_{x \neq y} |f(x) - f(y)| / |x - y|$.

## 4.2 Main Results

### 4.2.1 Quantitative Result

We evaluate our proposed DSA framework against existing methods utilize standard metrics. The **best** and second-best results are highlighted in Table 1 for the nuScenes and Argoverse datasets (with a 2-second observation window). Table 2 for Waymo (with a 1-second observation window) respectively. The results demonstrate that our method outperforms most existing approaches, achieving superior performance in 9 out of 13 evaluation metrics and ranking second in 3 others. Specifically, the best results over baseline datasets in Section 4.1 are 5.52%-$FDE_5$ (nuScenes), 8.82% -$ADE_6$ (Argoverse) and 1.93%-minFDE (Waymo).

Table 1: Performance comparison of baseline and our DSA framework on the nuScenes (left, N-Method) and Argoverse (right, A-Method) datasets. The **best** and second-best are highlighted.

| N-Method | $ADE_1$ | $FDE_1$ | $ADE_5$ | $FDE_5$ | $ADE_{10}$ | $FDE_{10}$ | A-Method | $ADE_1$ | $FDE_1$ | $ADE_6$ | $FDE_6$ |
|---|---|---|---|---|---|---|---|---|---|---|---|
| THOMAS [70] | - | 6.71 | 1.33 | - | 1.04 | - | GOHOME [71] | 1.70 | 3.68 | 0.89 | 1.29 |
| PreTraM [72] | - | - | 1.70 | 4.15 | 1.45 | 3.22 | LTP [13] | 1.62 | 3.55 | 0.83 | 1.30 |
| Goal-Driven [73] | - | - | 1.85 | 3.87 | 1.32 | 2.50 | MP++* [74] | 1.62 | 3.61 | 0.79 | 1.21 |
| MUSE-VAE [75] | - | - | 1.38 | 2.90 | 1.09 | 2.10 | HiVT [76] | 1.60 | 3.53 | 0.77 | 1.17 |
| Real-Time [77] | 3.56 | 8.63 | 1.60 | 3.34 | 1.23 | 2.32 | ADAPT [78] | 1.59 | 3.50 | 0.79 | 1.17 |
| Aware [79] | 5.58 | 11.47 | - | - | 1.67 | 2.66 | Aware [79] | 1.61 | 3.54 | 0.86 | 1.31 |
| FRM [80] | - | 6.59 | 1.18 | - | 0.88 | - | FRM [80] | - | - | 0.82 | 1.27 |
| Context-Aware [81] | 3.54 | 8.24 | 1.59 | 3.28 | - | - | R-Pred [82] | 1.58 | 3.47 | 0.76 | 1.12 |
| LAformer [11] | - | - | 1.19 | - | 0.93 | - | LAformer [11] | - | - | 0.77 | 1.16 |
| DAMM [83] | 2.84 | 6.59 | 1.39 | 3.14 | 1.02 | 2.05 | DAMM [83] | 1.57 | 3.42 | 0.76 | 1.29 |
| CASPNet++ [84] | 2.74 | **6.18** | 1.16 | 0.92 | - | | ProphNet [85] | 1.28 | **2.77** | 0.68 | 0.97 |
| CASPFormer [86] | - | 6.70 | **1.15** | - | - | - | QCNet [87] | - | - | 0.73 | 1.07 |
| DSA | **2.69** | 6.47 | 1.21 | **2.74** | **0.85** | **2.00** | DSA | **1.17** | 2.85 | **0.62** | **0.95** |

In the nuScenes dataset, DSA outperforms previous methods in metrics of $ADE_1$, $FDE_5$, $ADE_{10}$, and $FDE_{10}$. Compared to DAMM [83], which utilizes higher-order patterns to describe interactions between agents (vehicles), our model shows a significant improvement, achieving a 16.67% enhancement in $ADE_{10}$. Moreover, compared with FRM [80], which uses lane information to predict stochastic future relationships among agents, there is only a marginal gap of 0.03 in $ADE_5$, but we achieve a 3.41% improvement in $ADE_{10}$. However, DSA is slightly less effective than CASPNet++ [84], which employs interaction modeling and scene understanding for joint prediction of all road users while we only predict vehicle, that leads to a minimal gap, measured in the thousandth place.

In the Argoverse dataset, our model achieves on three of four metrics in baseline. Although our $FDE_1$ metric gap in 0.08 than the baseline best results ProphNet [85], when predicting 6 samples, DSA shows improvements of 8.82% in ADE and 2.06% in ADE. In addition, while ProphNet utilizes an agent-centric model with anchor informed strategies, our DSA employs global positioning directly.

In the Waymo dataset (Table 2), our DSA achieve the lowest minFDE and MR. We reduce the MR by 5.14% and minFDE by 1.88% compared to MotionLM [94], their minADE is slightly higher than ours by 0.015, whereas our framework is based on a simpler baseline model The ControlMTR [97] generate scene-compliant intention points and converte into a physics-based model, while DSA is driving and mathematical based, we reduce the minFDE with 4.10% (value 0.0488).

Table 2: Performance comparison of baseline and our DSA framework on the Waymo datasets. The **best** and second-best are highlighted.

| Method | minADE | minFDE | MR* |
|---|---|---|---|
| MultiPath++ [74] | 0.9780 | 2.3050 | 0.4400 |
| SceneTransformer [88] | 0.6117 | 1.2116 | 0.1564 |
| MPA [89] | 0.5913 | 1.2507 | 0.1603 |
| ReCoAt [90] | 0.7703 | 1.6668 | 0.2437 |
| DIPP [91] | 0.6951 | 1.4678 | 0.1854 |
| LiMTR [92] | 1.3640 | - | 0.2156 |
| HDGT [93] | 0.5933 | 1.2055 | 0.1511 |
| MotionLM [94] | **0.5702** | 1.1653 | 0.1327 |
| MTR++ [95] | 0.5912 | 1.1986 | 0.1296 |
| TC-Map [96] | 0.6181 | 1.2375 | 0.1402 |
| ControlMTR [97] | 0.5897 | 1.1916 | 0.1262 |
| DSA | 0.5852 | **1.1431** | **0.1259** |

* MR (Missing Rate) is the proportion of cases in which the Euclidean distance between the prediction and the ground truth at FDE exceeds 2m.

Our DSA framework adaptive design accommodates three categories of driving styles and we provide comprehensive explanations. This strategy simplifies the prediction process and enhances the accuracy and adaptability of predictions in complex real-world traffic scenarios.

#### 4.2.2 Qualitative Result

Figure 3 demonstrates the effectiveness for our DSA framework in vehicle trajectories prediction. For more visual content, please refer to Appendix. The $k$ is the number of generation trajectories, ground truth trajectory is actual trajectories. To describe specific subfigures in Figure 3, we use the

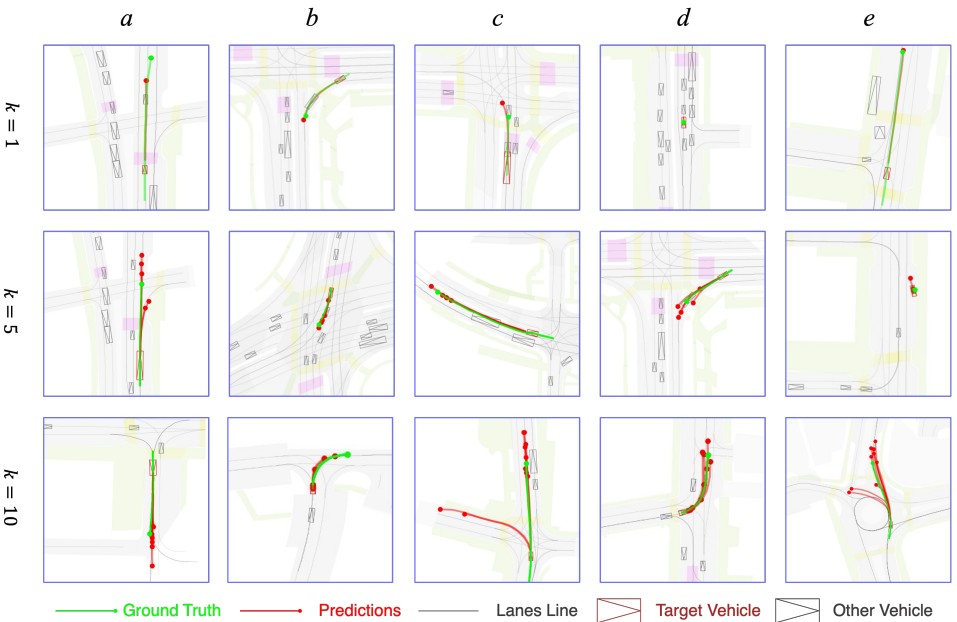

Figure 3: Qualitative results of our DSA framework. The value of $k$ (left) represents the number of generation trajectories while letters (top) are index for clearly describe. Round head lines represents predict and ground truth trajectory, respectively.

position index $(k, *)$ where $*$ denotes the letter shown at the top of each subfigure.

When $k = 1$ (i.e. the first row of Figure 3), he single prediction samples demonstrate that our DSA framework generally produces accurate results. It effectively handles not only simple road scenarios: such as straight lanes in $(1, a)$ and $(1, e)$, or stop conditions in $(1, d)$. But also complex scenarios including T-junctions in $(1, b)$ and crossroads in $(1, c)$.

In cases for generates 5 and 10 trajectories, our DSA framework delivers predictions that are both accurate and diverse. In simple scenarios, such as go straight in $(5, c), (10, a)$, or stopping in $(5, e)$, our framework maintains high accuracy while offering a broader range of plausible outcomes. It particularly excels in complex road conditions, including Y-crossroads in $(10, b)$, high density crossroads $(5, b)$ and roundabouts $(10, e)$. Moreover, the predicted trajectories effectively conform to curved roads, such as turning maneuvers in $(5, d)$ and $(10, d)$.

### 4.3 Ablation Studies for DSA Framework

To explore the benefits of different components and design choices in our DSA framework, we conduct ablation experiments along several dimensions: Type and combination of polynomials $p_n$ in Section 4.3.1. Degree adjustment of polynomials in Section 4.3.2. Analysis of driving style in Section 4.3.3, examining the relationship between styles and specific $p_n$, and how expert weights reflect behaviors. In addition, we present sensitivity analyses for different scenarios in Appendix D.

#### 4.3.1 Effects of Approximate Polynomial

We design two experiments to evaluate the combination and type of $p_n$ on the Argoverse dataset. **The number of polynomials.** To illustrate the importance of considering all drivers' driving styles instead of the parts of them in trajectory predictions, we simulate a scenario where only one or two driving styles existing and select corresponding matching $p_n$ (Section 3.2) for prediction.

Analyzing results from Table 3, consider whole driving style outperform almost the best than other combine. We observe that DSA framework incorporating two driving styles generally outperform those with only one. However, this trend is not universal. For instance, in $FDE_1$, the model

based solely on the normal driver (1.32) over the combination of conservative and aggressive styles (C+A, 1.54). Compared to the above combine (C+A) in $ADE_6$, DSA further re-

Table 3: The performances of DSA framework with different combinations of basis functions on the Argoverse dataset. C, A and N denote Conservative-$B_n$, Aggressive-$T_n^c$ and Normal-$L_n$, respectively. The **best** and second-best results are highlighted in table.

| Metric | C | A | N | C+A | A+N | C+N | DSA |
|---|---|---|---|---|---|---|---|
| $ADE_1$ | 1.61 | 1.45 | 1.32 | 1.54 | 1.39 | 1.66 | **1.17** |
| $FDE_1$ | 3.46 | 2.87 | 2.91 | 3.09 | 2.87 | **2.78** | 2.85 |
| $ADE_6$ | 1.02 | 1.31 | 1.12 | 0.83 | 0.92 | 0.98 | **0.62** |
| $FDE_6$ | 1.26 | 1.29 | 1.33 | 1.29 | 1.04 | 1.17 | **0.96** |

duces 0.21 with 25.3%. This illustrate that the necessity of considering all driving styles in trajectory prediction.

Table 4: Evaluate different $p_n$ in DSA framework on the Argoverse dataset, with original → replace and corresponding style (abbreviate with the first three letters) in the second column. The **best** and second-best results highlight in table.

| Method | Replace | $ADE_1$ | $FDE_1$ | $ADE_6$ | $FDE_6$ |
|---|---|---|---|---|---|
| $C_n + T_n + L_n$ | $B_n \to C_n$ Con | 1.48 | 3.76 | 1.10 | 1.53 |
| $B_n + S_n + L_n$ | $T_n \to S_n$ Agg | 1.59 | 3.91 | 0.86 | 1.41 |
| $B_n + T_n + H_n$ | $L_n \to H_n$ Nor | 1.62 | **2.81** | 0.71 | 1.02 |
| DSA | - | **1.17** | 2.85 | **0.62** | **0.96** |

**The Type of Polynomials.** For instruction the effects of the $p_n$ we utilize in DSA framework, we replace types of $p_n$ to evaluate it, with results in Table 4. We select Charlier ([98], $C_n$), Hermite ([99], $H_n$) and second-order ($S_n$) polynomials to instead $p_n$ we select in our DSA framework.

Our DSA yields the best performance on three out of four evaluation metrics in **blod**, which most improved is 27.8%, 43.6% and 37.3% respectively.

Although the combination $B_n + T_n + H_n$ achieves a slightly lower $FDE_1$ from 2.81 to 2.85 by 1.4% a marginal gap, DSA still ranks second on $FDE_1$.

### 4.3.2 Effects of Polynomial Degree

From Theorem 3.7, we understand that the prediction accuracy is directly related to the degree $n$ of the polynomial $p_n$. We now evaluate the impact of adaptively adjusting $n$. To clearly illustrate this influence, we analyze the performance of a single driving style with varying degrees, as shown in Figure 4. We observe that the error generally decreases with an increasing degree. However, the

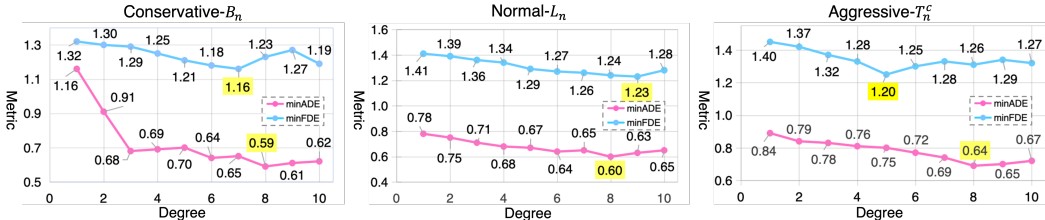

Figure 4: Performance of our DSA Framework,with only one single fixed basis function on the waymo dataset, which the lowest error highlighted in yellow.

highest degree does not necessarily yield the best results. For instance, within the aggressive driver style, $\partial (T_n^c) = 5$ outperforms other degrees in minFDE while $\partial (T_n^c) = 8$ is the best minADE, similar to conservative and normal driving style, the best results from different degrees. Consequently, adjusting $n$ rather than maintaining a fixed set provides enhanced granularity for the $p_n$ polynomials, thereby improving their capability to generate accurate and varied predictions across diverse vehicle trajectory styles.

### 4.3.3 Analysis of Driving Style

In Section 3.2, we provide the mathematical analysis for matching driving styles to their corresponding $p_n$. Here, we present experimental results that evaluate both the matching relationships and how expert weights reflect those relationships.

**Relationship between driving style and polynomials.** We compute the cosine similarity between the trajectory corresponding to the highest-weighted polynomial $p_n$ and the predefined driving style standards [48, 49, 50], which include Conservative (Con), Aggressive (Agg), and Normal (Nor). The results on the nuScenes (from N-Con) and Argoverse (from A-Con) datasets are shown in Table 5.

Table 5: Cosine similarity reflecting consistent matching between polynomial based predictions and driving style standards on the nuScenes (N) and Argoverse (A) datasets.

| $p_n$ | N-Con | Agg | Nor | A-Con | Agg | Nor |
|---|---|---|---|---|---|---|
| $B_n$ | **0.953** | 0.158 | 0.489 | **0.884** | 0.129 | 0.391 |
| $T_n$ | 0.266 | **0.987** | 0.353 | 0.167 | **0.927** | 0.329 |
| $L_n$ | 0.310 | 0.263 | **0.992** | 0.206 | 0.333 | **0.963** |

From Table 5, our matching scheme ($B_n$-Con, $T_n$-Agg, $L_n$-Nor) obtains significantly higher similarity scores on both datasets. On the nuScenes dataset, the average similarity of the three correct matches (diagram values) is 0.977, while that of all other matches (non-bold entries) is 0.307. On Argoverse, these values are 0.925 and 0.259, respectively, representing a substantial gap.

Table 6: The matching relationship between different driving styles and their corresponding $p_n$. The percentage indicates the rate at which each $p_n$ has the highest weight within each driving style.

| Style | Argoverse | nuScenes | Top-1 |
|---|---|---|---|
| Con | 86.15% | 87.96% | $B_n$ |
| Agg | 92.87% | 94.26% | $T_n$ |
| Nor | 80.04% | 83.56% | $L_n$ |

**Expert weights mapping to driving style.** To examine how expert weights mapping to driving styles, we compute the correct matching rate, defined as the proportion of cases in which the highest expert weight (Top-1) matches the expected $p_n$ for each driving style. The results appear in Table 6. The average correct matching rates across the two datasets are 86.35% and 88.59%, respectively. The highest matching rate is 94.26% for aggressive drivers on the nuScenes dataset. All styles achieve a matching rate above 80%, with a fluctuation range of 14.22%.

## 5 Limitation

We evaluate our method on prediction horizons up to 9 seconds (1 second of history and 8 seconds of future), which is the longest duration available in current open-access vehicle trajectory datasets. However, for significantly longer horizons (e.g., over one minute), direct long-term prediction may be unreliable and would likely require segment-wise modeling or hierarchical strategies. In addition, external factors such as strong conditions (e.g., traffic signals and regulatory constraints) and soft conditions (e.g., weather, which is often unlabeled in current datasets) can also affect trajectory prediction. Incorporating these contextual cues remains an important direction for future work.

## 6 Conclusion

We propose an adaptive framework for vehicle trajectory prediction that is tailored to the driving styles of human drivers. To enable effective matching between polynomials $p_n$ and driving styles, we analyze the behavioral characteristics of each style alongside the mathematical properties of corresponding $p_n$. Furthermore, we investigate the effects of $p_n$ combine, the influence of different polynomial types, and the necessity of adaptive parameters such as degree. Experiments results on three real-world datasets demonstrate that our framework significantly outperforms existing methods.

## Acknowledgements

This research is sponsored by the National Natural Science Foundation of China (U21B2090, 62472238, 62576181), the National Key Research and Development Program of China (2023YFB4301900), the Shenzhen Science and Technology Program (JCYJ20240813151445059), and the Science and Technology Planning Project of Guangdong Province (2023B12120600291).

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

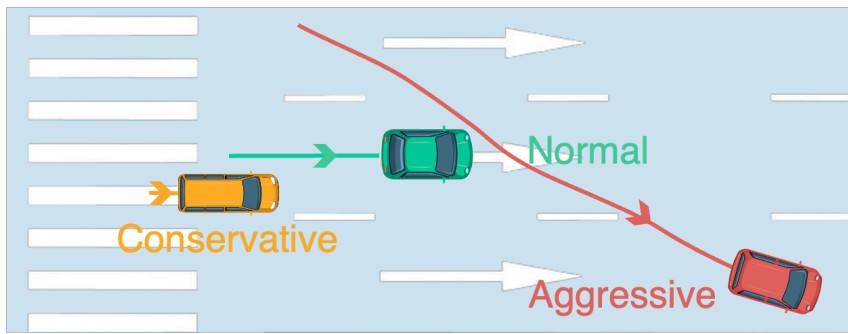

The diagram illustrates of three driving styles: Conservative, Aggressive and Normal (CAN), with their corresponding trajectories represented by lines recorded at each time step. The lengths of the lines indicate the driving distance, while the direction is shown by arrows. Conservative drivers typically move at low speeds or stop to avoid obstacles. Aggressive drivers often travel at high speeds and are prone to overtaking other vehicles. Normal drivers maintain a constant speed and frequently change lanes to ensure safety.


# Appendix of A Driving-Style-Adaptive Framework
# for Vehicle Trajectory Prediction

## A   Problem Background

### A.1   Vehicle Trajectory Prediction

The methods for vehicle trajectory prediction can be broadly classified into four categories [100]. These include: (i) Physics-based methods: These employ vehicle dynamics or kinematics models, such as singles trajectory methods, Monte Carlo, and Kalman filtering methods [101, 102, 103, 104, 105, 106]. These methods are known for their conciseness, efficiency, and computational effectiveness. (ii) Classic machine learning: Unlike physics-based methods that rely on several physics models, classic machine learning approaches apply data-driven models and consider additional factors for predicting trajectories. Examples include the Hidden Markov Model, Dynamic Bayesian Network, and K-Nearest Neighbors [107, 108, 109, 110, 111]. However, these traditional methods are typically only suitable for simple prediction scenarios and short-term prediction tasks.

Recently, with the advancement of modern machine learning, vehicle trajectory prediction methods based on: (iii) Deep learning and (iv) Reinforcement learning become increasingly popular. These methods are capable of considering interaction-related factors, understanding high-dimensional complex policies, and adapting to more complex scenarios. Examples include Graph Convolutional Network, Graph Attention Network, Conditional Variational Auto Encoder, and reinforcement learning techniques such as Inverse Reinforcement Learning, Generative Adversarial Imitation Learning, and Deep IRL [112, 113, 83, 114, 115, 116, 117].

In summary, an increasing number of autonomous vehicle trials are utilizing deep learning or reinforcement learning methods to predict future vehicle trajectories. These approaches leverage expert demonstrations and extract interaction information from traffic participants and road conditions, considering a broader range of influencing factors.

### A.2   Kolmogorov-Arnold Networks (KANs)

Hilbert's 13th problem [118] famously posits that it is impossible to solve general seventh-degree equations using only functions of two variables. Subsequent research by Kolmogorov et al.[36] has shown that any function involving multiple variables can be represented using a finite number of three-variable functions. Further studies as detailed by Arnol'd et al. [37], establish that even functions of just two variables are sufficient, as described in Theorem 2.1 presents significant for machine learning: learning a high-dimensional function essentially reduces to learning a limited number of one-dimensional basis functions $\psi(x)$ in Equation (2).

In reference [20], the authors introduce Kolmogorov-Arnold Networks (KANs), which are neural network applications based on Theorem 2.1. Unlike Multi-Layer Perceptrons (MLPs) that founded on the universal approximation theorem [119, 120, 121, 122], KANs feature learnable activation functions on what are traditionally referred to as "edges" (neurons) and they utilize fixed activation functions at what are typically called "nodes" (weights). Uniquely, each weight in KANs is replaced by a univariate function parametrized as a spline, meaning the network contains no linear weights whatsoever.

A variety of KANs are used across different tasks as noted in [123], such as solving ordinary differential equations [44, 124], image classification and reconstruction [125, 126], and time series forecasting [127, 128, 129], among others. These applications demonstrate competitive or superior performance in efficiency and predictive power compared to traditional models. However, to the best of our knowledge, we are the first to utilize KANs in vehicle trajectory prediction. This involves approximating and predicting trajectories for different driving styles, expanding the range of basis functions, and providing explanations for specific matches between functions and trajectories.

# B  Dataset Information

## B.1  Datasets

We preprocess the three datasets using the official pip packages provided by their respective baselines. The characteristics of each dataset are summarized in Table 7.

Table 7: Characteristics of the evaluation datasets.

| Datasets | Collect | Size | Library | Select |
|---|---|---|---|---|
| Argoverse [68] | Miami and Pittsburgh | 23.69G | Argoverse API / devkit | Motion Forecasting |
| nuScenes [67] | Boston and Singapore | 4.81G | nuscenes-devkit | Motional-Full dataset |
| Waymo [69] | Phoenix, AZ, Kirkland etc. | 83.50G | waymo-open-dataset | Motion1.1-scenario |

**nuScenes**  This dataset [67] offers high-definition maps and trajectory data from 1,000 driving scenes in Boston and Singapore, areas noted for dense traffic and complex driving challenges. It comprises 245,414 trajectory instances, each a sequence of 2D coordinates over 8 seconds, sampled at 2Hz. The nuScenes benchmark requires predicting a target agent's 6-second future trajectory from a 2-second historical trajectory. The comprehensive dataset features approximately 1.4 million camera images, 390,000 LIDAR sweeps, 1.4 million RADAR sweeps, and 1.4 million object bounding boxes across 40,000 keyframes.

**Argoverse**  This dataset [68] facilitates research in 3D tracking and motion forecasting for autonomous vehicles. Originating from select areas in Miami and Pittsburgh, it includes 113 scenes with 3D tracking annotations, featuring 324,557 significant vehicle trajectories derived from over 1,000 hours of driving. The forecasting component of Argoverse provides agent trajectories and high-definition maps, requiring the prediction of a target vehicle's future trajectory for the next 3 seconds, based on its past trajectory over two seconds, sampled at 10Hz. The dataset encompasses 333K real-world driving sequences, primarily at intersections or within dense traffic, each focusing on one target vehicle for trajectory prediction.

**Waymo**  This dataset [69] publicly to aid the research community in investigating a wide range of interesting aspects of machine perception and autonomous driving technology. This Dataset we use is the Motion part, with object trajectories and corresponding 3D maps for 103,354 segments. Given agents' tracks for the past 1 second on a corresponding map, predict the joint future positions of 2 interacting agents for 8 seconds into the future. The ground truth future data for the interactive test set is hidden from challenge participants. The validation sets contain the ground truth future data for use in model development. In addition, the test and validation sets provide 2 interacting object tracks in the scene to be predicted.

## B.2  Metrics

We evaluate the predicted trajectory $Y^t$ against the ground truth trajectory $Y_{GT}^t$ using standard error-based metrics. Our DSA framework adopts the commonly used Average Displacement Error (ADE) and Final Displacement Error (FDE), as defined in [23]:

$$\text{ADE} = \frac{1}{T}\sum_{t=1}^{T}\|Y^t - Y_{GT}^t\|_{L^2}, \ \text{FDE} = \|Y^t - Y_{GT}^t\|_{L^2}.$$

Here, the superscript $t$ denotes the current time step, and $T$ refers to the total number of time steps in the prediction horizon. The metrics we use include $\text{ADE}_k$, $\text{FDE}_k$, minADE, minFDE, and b-minFDE. The subscript $k$ indicates the Top-$k$ most likely predicted future trajectories. The "min" variants (minADE and minFDE) compute the $L^2$ distance between $Y_{GT}^t$ and the closest predicted trajectory $Y^t$ across all generated samples, averaged over all agents. The b-minFDE metric extends minFDE by incorporating the Brier score [130], which evaluates the calibration of the predictive distribution. It is defined as the sum of the Brier score and minFDE.

# C  Proof of Bound

As we discuss in Section 3.3.2, Kolmogorov's theorem [131] provides error bounds by evaluating the absolute value of the function and the overall variation in the function value. This is illustrated as follows:

**Theorem C.1** (***Kolmogorov Theorem***) *For $f \in C\,[a, b]$, there exists a polynomial $p_n$ such that approximation error is bounded by:*

$$\|f - p_n\|_{L^\infty} \lesssim \left( \frac{\log n}{n} \right) V(f, [a, b]),$$

*where $V(f, [a, b])$ denotes the total variation* [9] *of $f$ over the interval.*

From this theorem, we conclude that when $n > e$, increasing the degree $n$ of $p_n$ results in a smaller decrease in the theoretical upper bound on the approximation error.

However, in practice, considering computational cost and time, the value of $n$ cannot be arbitrarily large. In this section, we provide three proofs corresponding to the three categories of drivers, demonstrating that under a given error limit $\delta$, there exists a relationship between the minimum degree $n$ and the components of the trajectory to be approximated, as described in Section 2.1 (i.e., position $(x, y)$, velocity, and acceleration). For clarity, we use $f$ to represent each continuous component with respect to $t$.

## C.1  Conservative Drives: Bernstein Polynomial

In Section 3.2.1, we use the properties of Bernstein polynomials $(B_n)$ for their uniform convergence to approximate the trajectories of conservative drivers, characterized by low speed and minimal motion changes. The minimum degree $n$ of the $B_n$ polynomial is obtained by:

**Theorem C.2** *For all $\epsilon > 0$, if $\partial\,[B_n\,(f)] = n$, then for a given error limit $\delta$ with $0 < \epsilon \leqslant \delta \ll \infty$, then*

$$n \geqslant max \mid f''\,(\xi) \mid /8\delta.$$

*Proof:*  First, we calculate the error between $f(x)$ and $B_n(f)$

$$|f\,(x) - B_n\,(f)|$$

$$= \left| \sum_{k=0}^{n} \left[ f\,(x) - f\left(\frac{k}{n}\right) \right] \binom{n}{k} x^k\,(1 - x)^{n-k} \right|$$

$$\leqslant \left| \sum_{k=0}^{n} \left[ -f''^{\left(\frac{k}{n} - x\right)} - \frac{1}{2} f''\left(\frac{k}{n} - x\right)^2 \right] \cdot P_B\,(k) \right| \tag{4}$$

$$= \frac{1}{2} f''\,(\xi) \sum_{k=0}^{n} \left(\frac{n}{k} - x\right)^2 P_B\,(k) \qquad \xi \in \left(x, \frac{k}{n}\right). \tag{5}$$

In formula (4), $P_B\,(k) \triangleq C_n^k x^k\,(1 - x)^{n-k}$. From Equation (5), we next proceed to prove

1. $\sum_k \left(\frac{n}{k} - x\right) \cdot P_B\,(k) = 0$,
2. $\sum_k \left(\frac{n}{k} - x\right)^2 \cdot P_B\,(k) = \frac{x}{n}\,(1 - x)$.

For Equation 1, applying the Central Limit Theorem (CLT) as discussed in [132], we consider the total difference of weights, represented by $[(n/k) - x]$. Specifically, the probability $p$ satisfies $p = x = k/n$. In this case, as described in [133], we have $E(k) = nx$. Thus, we can derive the expectation as follows:

$$E\left(\frac{k}{n}\right) = \frac{E\,(k)}{n} = x. \tag{6}$$

Therefore, Equation 1 simplifies to $E(k/n) - x = 0$.

For Equation 2, we employ a similar method; here $\left(\frac{n}{k} - x\right)^2$ represents the squared difference in weights between $\frac{k}{n}$ and $x$, alternatively described as the deviation between observation and expectation. According to Equation (6),

$$D\left(\frac{k}{n}\right) = \frac{nx}{n^2} \cdot (1 - x) = \frac{x\,(1 - x)}{n}.$$

---

[9]**Total Variation** A measure of the total amount of variation in a function over a given interval $[a, b]$, which is defined by $\sup_{x \neq y} |f\,(x) - f\,(y)| \,/\, |x - y|$.

Equation 2 corresponds to the squared deviation $\left(\frac{n}{k} - x\right)^2$ based on weights $P_B(k)$. Moreover

$$\because E\left[\left(\frac{k}{n} - x\right)^2\right] = D\left(\frac{k}{n}\right),$$

$$\therefore (5) = \frac{1}{2} f''(\xi) \frac{x(1-x)}{n} \leqslant \frac{M_2}{8n}, \text{ with } M_2 = \max_{\xi \in \mathbb{D}} f''(\xi).$$

Considering the error limit $\delta$, we have:

$$\frac{M_2}{8n} < \delta \Rightarrow n \geqslant \frac{M_2}{8\delta}.$$

$\square$

**Theorem C.3** *If $f \in L^p[a,b]$, $B_n^\omega(f) \in [a,b]$ and $\partial[B_n^\omega(f)] = n$. For all $\epsilon > 0$, $0 < \epsilon \leqslant \delta \ll \infty$, $\delta$ is given error limitation, then:*

$$n \geqslant \frac{\max[|f''(\widetilde{\xi}) \cdot \omega|] \cdot (b-a)^2}{8\delta},$$

*where $\omega$ is the weights of weighted $B_n$ polynomials $B_n^\omega(f)$.*

*Proof:* Here the error is $L^\infty$ norm, the definition of $B_n^\omega(f)$ is

$$B_n^\omega(f) = \sum_{k=0}^{n} f\left(\frac{k}{n}\right) \omega\left(\frac{k}{n}\right) \binom{n}{k} x^k (1-x)^{n-k}$$

Let $u = (x-a)/(b-a)$, then $u \in [0,1]$. So $\widetilde{M_2}$ is similar to Theorem C.2, here

$$\widetilde{M_2} = \max_{u \in [0,1]} |g''(u)| = (b-a)^2 |f''(\xi)|,$$

where $g(u) = f\left(\frac{u-a}{b-a}\right)$. The next following prove is similar to Theorem C.2. $\square$

## C.2 Aggressive Drivers: Chebyshev Polynomial

In Section 3.2.2, aggressive drivers' trajectories are characterized by non-smooth, high-speed movements during motion changes. We use the Chebyshev polynomials $T_n^c$ and their minimum-maximum error properties to approximate these trajectories. The minimum degree $n$ of $T_n^c$ polynomial is obtained as follows:

**Theorem C.4** *For $f \in L^p[a,b]$ and a given error bound $\delta$ (where $0 < \epsilon \leqslant \delta \ll \infty$), the condition $\partial[T_n^c(f)] = n$ is satisfied:*

$$n \geqslant \frac{1}{\omega^{-1}\left(f, \frac{\delta}{12}\right)}, \tag{7}$$

*where $\omega^{-1}$ is the inverse of the modulus of continuity for the function $f$.*

To provide the proof of Theorem C.4, we first introduce the definition of the modulus of continuity and a lemma related to this proof.

**Definition C.5** *(Modulus of Continuity in $L^p$ Space) Let $f \in L^p[a,b]$, $p \geqslant 1$ and $0 \leqslant m \leqslant b-a$. The modulus of continuity $\omega_p(m, f)$ is defined as:*

$$\omega_p(m) = \omega_p(m, f)$$

$$= \sup_{0 \leqslant h \leqslant m} \left(\int_a^{b-h} |f(x+h) - f(x)|^p\right)^{1/p}$$

*which represents the continuity norm for $f$ over the interval $[a,b]$.*

For $T_n^c$ polynomials belonging to the $C_{2\pi}$ space [10], we use $E_n(f)$ to denote the deviation of the approximation of $f$ by a trigonometric polynomial $T_n$ of degree $n$, as follow:

$$E_n(f) = \inf_{\{T_n\}} \|f - T_n\|.$$

This deviation satisfies:

---

[10]$C_{2\pi}$ **Space** Let $f \in \mathbb{R}$ with period $2\pi$. Define

$$\| f \| = \sup_{-\pi \leqslant x \leqslant \pi} |f(x)|.$$

We call the above set the $C_{2\pi}$ space

**Lemma C.6** (*Jackson [134]*) Let $f \in C_{2\pi}$, then for all $n \in \mathbb{N}$, the following inequality holds:

$$E_n(f) \leqslant 12 \cdot \omega \left( f, \frac{1}{n} \right).$$

It is evident that $T_n^c \subseteq C_{2\pi}$. Based on Lemma C.6, we present the proof of Theorem C.4.

*Proof:* According to the definition of $\delta$, we have $\| f - T_n \|_{L^\infty} < \delta$. Lemma C.6 provides the modulus of continuity under the $L^\infty$ space, so we need to relate $\omega_p$ from Definition C.5 to $\omega$ in Lemma C.6, which relates the $L^p$ and $L^\infty$ norms:

$$\|g\|_{L^p([a,b-h])} \leq (b - a - h)^{\frac{1}{p}} \cdot \|g\|_{L^\infty([a,b-h])}.$$

For the function difference $g(x) = f(x + h) - f(x)$:

$$|f(x + h) - f(x)| \leq \omega(f, h), \text{ for } \forall x \in [a, b - h].$$

Therefore, the $\omega_p(m, f)$ is related to $\omega(f, h)$ as follows:

$$\omega_p(h, f) \leqslant \left\{ \int_a^{b-h} [\omega(f, h)]^p \right\}^{1/p}$$

$$= \omega(f, h) \cdot (b - a - h)^{\frac{1}{p}}. \tag{8}$$

When $h \to 0$, Equation (8) can be approximated as:

$$\omega_p \leqslant \omega \cdot (b - a)^{\frac{1}{p}}.$$

According to Lemma C.6 and satisfy the error limit $\delta$, s.t. $E_n(f) \leqslant \delta$, we have

$$\omega \left( f, \frac{1}{n} \right) \leqslant \frac{\delta}{12}.$$

To obtain the lower bound on $n$, since $\omega(f, h)$ is a nondecreasing function with respect to $h$, we take its inverse function $\omega^{-1}(f, y)$ as follows:

$$\frac{1}{n} \leqslant \omega^{-1} \left( f, \frac{\delta}{12} \right). \tag{9}$$

Finally, the lower bound for $n$ can be derived from inequality (9) as:

$$n \geqslant \frac{1}{\omega^{-1} \left( f, \frac{\delta}{12} \right)}.$$

$\square$

Lemma C.4 provides a minimum bound related to the value of the modulus of continuity. Furthermore, the Lipschitz continuity [135] of vehicle trajectories $X_i$ and $Y_i$ offers a more compact bound for inequality (7):

**Corollary C.7** *The bound in inequality (7) is satisfied as follows:*

$$n \geqslant \frac{12L}{\delta}, \tag{10}$$

*where $L$ represents Lipschitz constant.*

*Proof:* The proof of Corollary C.7 consists of two parts: (i) establishing Lipschitz continuity of vehicle trajectories and (ii) deriving Equation (10).

(i) Lipschitz Continuity of Vehicle Trajectories

To demonstrate the Lipschitz continuity of vehicle trajectories, it suffices to show that their state information (Section 2.1), including $(x, y)$ position, velocity $v$ and acceleration $a$, satisfies the Lipschitz condition ($L$-condition). Specifically, there exists a constant $L$, s.t. for any $x', x'' \in [a, b]$, the following holds:

$$\left| f(x') - f(x'') \right| \leqslant L \left| x' - x'' \right|. \tag{11}$$

According to the physical relationships among these states, if acceleration $a$ satisfies $L$-condition, then by the boundedness theorem [136], the other states also satisfy it. Thus, we take $a$ as an example, and similar arguments apply to the other states, quod erat demonstrandum.

According to [137, 138], the variation in vehicle acceleration is constrained by factors such as engine performance, vehicle weight, and braking system, which implies that the jerk $j(t)$ (the rate of change of acceleration over time)

cannot be physically infinite. Therefore, there exists a constant $M_j$, s.t. $|j(\tau)| \leqslant M_j$. For any $t_1, t_2 \in [a, b]$, the following holds:

$$|a(t_1) - a(t_1)| = \left| \int_{t_2}^{t_1} j(\tau) \right| \leqslant \int_{t_2}^{t_1} |j(\tau)|$$

$$\leqslant \int_{t_2}^{t_1} M_j \, d\tau = M_j |t_1 - t_2|.$$

Therefore,Hence, the acceleration function $a(t)$ is Lipschitz continuous with the Lipschitz constant $L = M_j$.

(ii) Derivation of a more compact bound.

To obtain a tighter bound, we use the $L$-continuity property of $f$. From the continues of modulus, we have:

$$\omega(f, h) \leqslant Lh \Rightarrow \omega^{-1}(f, y) \geqslant \frac{y}{L}. \tag{12}$$

From Jackson's inequality in Lemma C.6, let $y = \delta/12$ in Equation (12). This ensures that the error remains below $\delta$, with $\delta/12$ acting as a piecewise error threshold. Then Equation (12) becomes:

$$\omega^{-1}(f, y) \sim \omega^{-1}(f, \frac{\delta}{12}) \leqslant \frac{\delta}{12L}. \tag{13}$$

Combining inequalities (7) and (13), we obtain the more compact bound (10). $\qquad \square$

## C.3 Normal Drivers: Legendre Polynomial

In Section 3.2.3, we discuss that the speed and acceleration of normal drivers maintain an intermediate state between conservative and aggressive drivers. Their trajectories do not change as dramatically as aggressive drivers, nor are they so slow as to affect the flow of traffic. To approximate these trajectories, we use the Legendre polynomial $L_n$. The minimum degree $n$ of $L_n$ is obtain by:

**Theorem C.8** *For all $\epsilon > 0$, if $\partial[L_n(f)] = n$, for a given error limit $\delta$ with $0 < \epsilon \leqslant \delta \ll \infty$, then*

$$n \geqslant \left( \frac{C_H}{\delta} \right)^{1/\alpha}, \tag{14}$$

*where $C_H$ is Hölder constant modulus of continuity and $\alpha$ represents Hölder exponent.*

From Theorem C.8 we apply Jackson's inequality (Lemma C.6) to establish a relationship between the approximation and the modulus of continuity. This inequality also applies to continuous functions defined on the interval $[-1, 1]$ interval. To achieve the bound in Equation (14), we further use the Hölder continuous property [139] of vehicle trajectory. Similar to $L$-continues as defined in Equation (11), Hölder continuous (H-continuous) is define as follows:

**Definition C.9** *(Hölder continuous) For a function $f$ defines on interval $I$, if there exits a constant $C \in \mathbb{R}$, s.t. for $\forall z', z'' \in I$:*

$$\left| f(z') - f(z'') \right| \leqslant C_H |z' - z''|^\alpha, \ \alpha \in (0, 1],$$

*then $f$ is is said to be Hölder continuous of order $\alpha$.*

When $\alpha = 1$, $H$-continuous reduces to $L$-continuous. Reference [140, 141, 142, 143] analyze the Hölder continuity or related smoothness of vehicle trajectories and their states (position, velocity, and acceleration), either directly or indirectly by examining the smoothness of physical constraints and changes. Therefore, the bound in Equation (14) can similarly be derived from inequality (12):

$$\omega(f, h) \leqslant C_H h^\alpha \Rightarrow C_H \left( \frac{1}{n} \right)^\alpha \leqslant \delta.$$

Thus, we obtain a bound as expressed by inequality (14).

# D Effects of Framework Sensitive

In Section 4.3, we evaluate our DSA framework with respect to module component dimensions: namely the type, combination, and degree of polynomials $p_n$, as well as the driving style matching. In this section, we further analyze the sensitivity of our model with regard to: variations in the driving style categories themselves (Section D.1), external influences such as changing traffic densities (Section D.2) and varying road conditions (Section D.3).

## D.1 Number of Driving Style

In reference [48, 49, 50] we know that driving style are categores as three type and each characters are illustrated in Section 3.2. Now we use automatic manner category numbers rather than the predefining for driving styles. Specifically, we evaluate multiple-category settings using K-means clustering, and report the corresponding metrics (log-normalized results except for Silhouette metric) on the Argoverse (from A-$\triangle$WSS) and nuScenes (from N-$\triangle$WSS) in Table 8, here for all metrics listed below, larger values indicate better clustering performance. Our three-category configuration yields the highest number of best scores across the metrics, supporting the

Table 8: Clustering evaluation results on Argoverse (A) and nuScenes (N). A value of "1" indicates the best performance among all settings.

| $k$ | A-$\triangle$WSS | Silhouette | $\triangle$CHI | N-$\triangle$WSS | Silhouette | $\triangle$CHI |
|---|---|---|---|---|---|---|
| 2 | - | 0.919 | - | - | 0 | - |
| 3 | **1** | **1** | 0.810 | **1** | 0.700 | **1** |
| 4 | 0.743 | 0.986 | **1** | 0.586 | **1** | 0.387 |
| 5 | 0.022 | 0.283 | 0 | 0.287 | 0.268 | 0.324 |
| 6 | 0 | 0 | 0.601 | 0 | 0.398 | 0 |

validity and rationality of our chosen driving style taxonomy. The metrics [144] are defined as follows:

- WSS (Within-Cluster Sum of Squares): Measures the improvement in intra-cluster compactness. A higher value suggests tighter grouping of samples within clusters after clustering.

- Silhouette: Reflects both the cohesion within a cluster and the separation between clusters. A higher silhouette score indicates that samples are well matched to their own cluster and poorly matched to neighboring clusters.

- CHI (Calinski-Harabasz Index): Captures the variation in inter-cluster separability and intra-cluster compactness. Higher values indicate better-defined and more distinct clusters.

## D.2 Traffic Density

Traffic density is closely related to vehicle speed and traffic flow, and significantly influences trajectory prediction due to varying interaction patterns among vehicles [145]. To clearly present the impact of traffic density, we divide the dataset into five levels based on the number of vehicles per unit area. We compare the performance of our DSA framework against the best-performing baseline with publicly available code: Context-Aware [81], as identified in Table 1. The comparison results are illustrated in Figure 5.

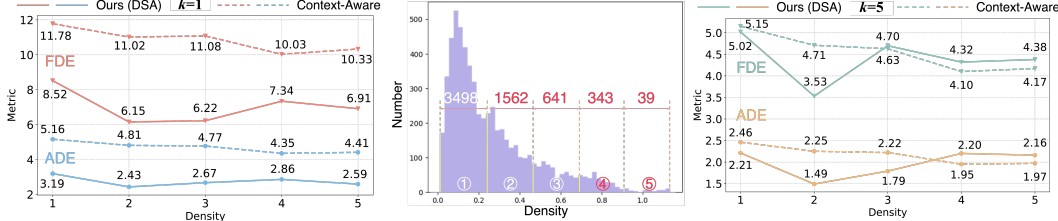

Figure 5: Comparison of trajectory prediction performance under different traffic densities with Context-Aware [81]. Our method is represented by solid lines, while Context-Aware is depicted using dashed lines on both sides. The middle subfigure shows the density distribution with circled numbers indicating the corresponding density levels.

Our DSA framework consistently outperforms the baseline across over 75% of the dataset. When averaged over the highest-density and most common case (density level 1 in the middle subfigure of Figure 5), our method achieves improvements of 1.97 in $ADE_1$ and 3.26 in $FDE_1$, as well as 0.25 in $ADE_5$ and 0.13 in $FDE_5$.

The most notable gains appear in density level 2, where DSA reduces prediction errors by 49.48% and 44.19% for $k = 1$, and by 33.78% and 25.05% for $k = 5$, compared to Context-Aware. Although our framework shows slightly lower performance in levels 4 and 5 when generating 5 trajectories, these cases together account for only 6.28% of the dataset.

## D.3 Road Condition

Road structure significantly influences the motion patterns of agents navigating through urban or highway environments and is thus essential for accurate trajectory prediction [146, 11, 112]. Does road complexity increase the frequency of driving style changes, thereby making prediction more difficult? To investigate this question, we evaluate the performance of our DSA framework under different road conditions, as shown in Table 9.

Table 9: The b-minFDE* in different road condition on the nuScenes dataset. The **best** results are highlighted.

| Method Type | Stationary | Straight | Straight right | Straight left | Right U-turn | Right-turn | Left U-turn | Left-turn | All |
|---|---|---|---|---|---|---|---|---|---|
| MTR [147] | 2.15 | 2.58 | **4.85** | 4.26 | 8.13 | 4.82 | **5.17** | **4.85** | 2.86 |
| DSA | **2.03** | **2.48** | 4.96 | **4.17** | **8.11** | **4.77** | 5.28 | 4.87 | **2.75** |

* Brier-minFDE ($b$-minFDE): $b\text{minFDE}=\text{minFDE}+(1-p)^2$, where $p$ is the probability of probability of the best forecasting trajectory with minimum endpoint error (minFDE).

Our DSA framework achieves the lowest overall error of 2.75, improving upon the baseline by 3.85%. It outperforms the baseline in 6 out of 8 categories, including reductions in error for common scenarios such as Stationary (5.6%) and Straight (3.9%), as well as complex maneuvers like Straight-Left (2.1%), Right U-turn (0.25%), and Right Turn (1.0%).

Although MTR performs slightly better in Straight-Right and Left U-turn (by 0.11 in both cases), DSA matches or surpasses baseline performance in the most frequent and safety-critical trajectory types. These results demonstrate the robustness and adaptability of our framework across diverse road semantics, particularly in non-linear or discontinuous motion patterns.

# E  Detailed Description of the Algorithm

In Sections 3.3.1 and 3.3.2, we introduced the mechanisms for polynomial combination and degree adaptation. In this section, we provide a detailed description of the corresponding algorithms.

## E.1  Polynomial Combination

To match each trajectory under various driving styles to a suitable polynomial combination, as analyzed in Section 3.2, we employ a Mixture of Experts model based on Top-$K$ Gating (MoE-TopK) [63]. In this method, tunable Gaussian noise is added to the gating logits, and only the top $K$ values are retained for expert selection.

Let us denote by $G(x)$ and $E_j(x)$ the output of the gating network and the output of the $j$-th expert network for a given trajectory $X_i$, for clearly we omit subscript $i$. The output $z^{\mathrm{com}}$ of the MoE module can be written as follows:

$$z^{\mathrm{com}} = \sum\nolimits_{j=1}^{3} G(X)_j E_j(x). \tag{15}$$

As shown in [21, 61, 62], kernel density estimation and latent variable analysis reveal that a driver's behavior evolves continuously across different situations. This implies that driving behavior can be viewed as a probabilistic mixture of weighted driving styles. Given that drivers may exhibit behaviors characteristic of multiple styles in dynamic scenes—such as when another agent suddenly appears—we adopt the Noisy Top-$K$ Gating network [63] to capture this mixture behavior. This network activates only the top-$k$ best-matching experts, enhancing responsiveness and specificity. Accordingly, this modification adjusts $G_*(\cdot)$ in Equation (15) to $\widetilde{G(x)}$, detailed as follows:

$$\widetilde{G_j(x)} = \mathrm{Softmax}\left\{\mathrm{KeepTopK}\left[H(X), k\right]\right\}, \text{ with}$$

$$H(X) = (X \cdot W_{G_j}) + \mathrm{SN}() \cdot \mathrm{Sp}\left[\left(X \cdot W_{\mathrm{noise}_j}\right)\right].$$

Here, "SN" and "Sp" denote Standard Normal [64] and Softplus [65] functions, respectively. The symbol $W_*$ denotes the weight matrix corresponding to each subscript. The loss function is defined as follows:

$$L_{\mathrm{MoE\text{-}K}} = w_{\mathrm{load}} \cdot \mathrm{CV}\left(\mathrm{loads}\right)^2, \tag{16}$$

where "load" refers to the importance values assigned to each driving style, with $w_{\mathrm{load}}$ representing the corresponding weight. "CV" stands for the Coefficient of Variation, which assesses the variability of these values. Equation (16) is a part of Loss in Section 4.1. This structure of the MoE model effectively recognizes the diversity of trajectories, allowing each expert to specialize in different features of driving styles.

## E.2  Degree Adaptation

From Theorem 3.7, we understand that the accuracy of polynomial $p_n$ approximation is directly influenced by the degree $n$ of the $p_n$. However, adapting the degree of $p_n$ poses a complex, non-convex, and combinatorial optimization challenge, as the relationship between prediction error and polynomial degree is not straightforward. This complexity often leads to the presence of multiple local optima.

To address this issue, we utilize the versatile Bayesian Optimization (BO) tool SMAC3 [66] for its robustness and flexibility, making it particularly suitable for optimizing low-dimensional and continuous functions (type: SMAC4BB), such as those found in vehicle trajectory prediction.

We treat the adaptive of polynomial degree as a hyperparameter optimization problem, using SMAC3 for BO, which leveraging Gaussian Processes with the Matérn kernel and the Expected Improvement acquisition function, iteratively searches the candidate degree set to minimize the loss function. Specifically, the degree $n$ is treated as a hyperparameter optimization problem, aimed at minimizing the loss on validation data $D_{\mathrm{val}}$ of our model trained on training data $D_{\mathrm{train}}$. This process can be formulated as follows:

$$n_{\mathrm{SMAC}} \in \arg\min_{n \in \mathbb{Z}^+} c(n) = \arg\min_{n \in \mathbb{Z}^+} L\left(\mathcal{D}_{\mathrm{train}}, \mathcal{D}_{\mathrm{val}}; n\right),$$

The hyperparameter optimization process targets the degree $n_{\mathrm{SMAC}}$, which is defined as the optimal degree that achieves the least error for the corresponding basis function $p_n$. Here $L$ denotes the loss function.

# F   Limitation and Discussion

**Limitation**   We summarize existing open vehicle trajectory prediction datasets in Table 10, and observe that the maximum available trajectory duration is typically less than 10 seconds. Despite this limited time span, our framework based on three driving styles adapts well to such settings. We evaluate its performance on both short-term (3 seconds, Table 1) and long-term (8 seconds, Table 2) prediction tasks, achieving consistently strong or state-of-the-art results across all durations.

Table 10: Existing vehicle trajectory datasets. "His" and "Pre" represent the historical and predicted trajectory durations, respectively, while "Total" denotes the overall duration for each vehicle.

| Datasets | Pub. | Collect Locations | His | Pre | Total |
|---|---|---|---|---|---|
| KITTI [148] | 2012 CVPR | Karlsruhe | 2 | 4 | 6 |
| Apolloscapes [149] | 2018 CVPR | Beijing, ShangHai and SHenZhen | 3 | 3 | 6 |
| nuScenes [67] | 2020 CVPR | Boston and Singapore | 2 | 6 | 8 |
| Argoverse [68] | 2019 CVPR | Miami and Pittsburgh | 2 | 3 | 5 |
| INTERACTION [150] | 2019 arXiv | China, Germany and Bulgaria | 1 | 3 | 4 |
| InD [151] | 2020 TIV | German | 3.2 | 4.8 | 8 |
| RounD [152] | 2020 ITSC | Aachen | 2 | 4 | 6 |
| HighD [153] | 2018 ITSC | German | 2.8 | 2.8 | 5.6 |
| Waymo [69] | 2020 CVPR | USA | 1 | 8 | 9 |

However, in longer prediction horizons, the complexity of driving behavior increases, suggesting that three driving style categories may be insufficient to cover all possible scenarios. Moreover, trajectory patterns are often influenced by external factors, which can be categorized as either soft or strong conditions.

Soft conditions, such as weather, affect driver perception and reaction. For example, on sunny days, improved visibility may enhance drivers' responsiveness, leading to smoother and more stable trajectories. In contrast, adverse weather conditions such as fog, heavy rain, or snow can result in more abrupt or irregular driving behavior.

Similarly, strong conditions such as traffic signals or regulatory constraints also significantly influence vehicle trajectories. Unfortunately, most existing datasets lack labels for these contextual factors. We believe that incorporating such labels could further enhance prediction accuracy in future research.

**Discussion**   For longer vehicle trajectories, we can improve our DSA framework from both practical and theoretical perspectives.

1. Incorporating more driving styles. Our current DSA framework utilizes three representative styles: Conservative, Aggressive and Normal (CAN), which reflect two behavioral extremes and an intermediate pattern. However, as the temporal length of each driver's trajectory increases, driving behaviors may exhibit greater variability. To capture these nuances, the model can be extended by defining or integrating additional driving styles. This would allow for a more fine-grained characterization of driver behavior and potentially lead to improved trajectory prediction accuracy.

2. Expanding the set of basis functions. As driving styles become more diverse and trajectory conditions more complex, a broader set of basis functions is required to effectively approximate and predict vehicle trajectories. Instead of relying on a single polynomial type, we can extend to a set of basis functions of the same class, such as orthogonal trigonometric polynomials. For example, to minimize the $L^2$ norm in modeling trajectories of normal drivers, or other intermediate states between conservative and aggressive behavior. It is beneficial to use a richer set of orthogonal polynomials that better match the dynamics of these nuanced driving styles.

# G  Future Work

In this paper, we focus on the characteristics of the individuals who generate the data (i.e., trajectories) and leverage the mathematical properties of basis functions to approximate these trajectories. This concept can be generalized and extended to other domains, such as:

- Other traffic participants. In addition to drivers, other agents in the traffic scene such as pedestrians and cyclists, also exhibit distinct behavioral characteristics. By modeling these characteristics, we can select appropriate basis functions tailored to each agent type, thereby improving the accuracy of their trajectory prediction.
- Multivariety Time Series Forecasting. Our framework can be extended to long-term forecasting tasks in domains such as weather prediction, energy consumption and electrocardiography. For example, one could model temperature and precipitation trends across different climate zones, analyze electricity usage patterns based on consumer behavior, or study heart rate dynamics as a function of individual health conditions.

Additionally, by leveraging the core theoretical foundation (Theorem 2.2), we aim to construct models grounded in the physical characteristics or behavioral attributes of the data sources, thereby fully exploiting the inherent structure of the data itself.

# H  Visualization

Due to space constraints, the number of visualized prediction results in the main text is limited. Here, we provide additional visualizations of predicted trajectories for various scenes, with generated trajectories $k = 1, 5, 10$. Each value of $k$ is presented for both simple (e.g., straight roads) and complex scenes (e.g., turns conditions), showcasing different types of driving behavior.

To enhance the clarity of the visualization results, we present them on a dedicated page and reduce the background opacity to improve visual contrast. Specific outcomes are accompanied by detailed explanations provided in the corresponding figure captions. In summary, considering various scenario combinations and adjusting the number of generated trajectories lead to more diverse, accurate, and comprehensive vehicle trajectory predictions. Increasing the number of predicted trajectories improves prediction diversity and realism, while analyzing different scenarios helps adapt to the diversity and complexity of real-world traffic environments. These improvements contribute to making the model both more mathematically grounded and more adaptive.

- $k = 1$

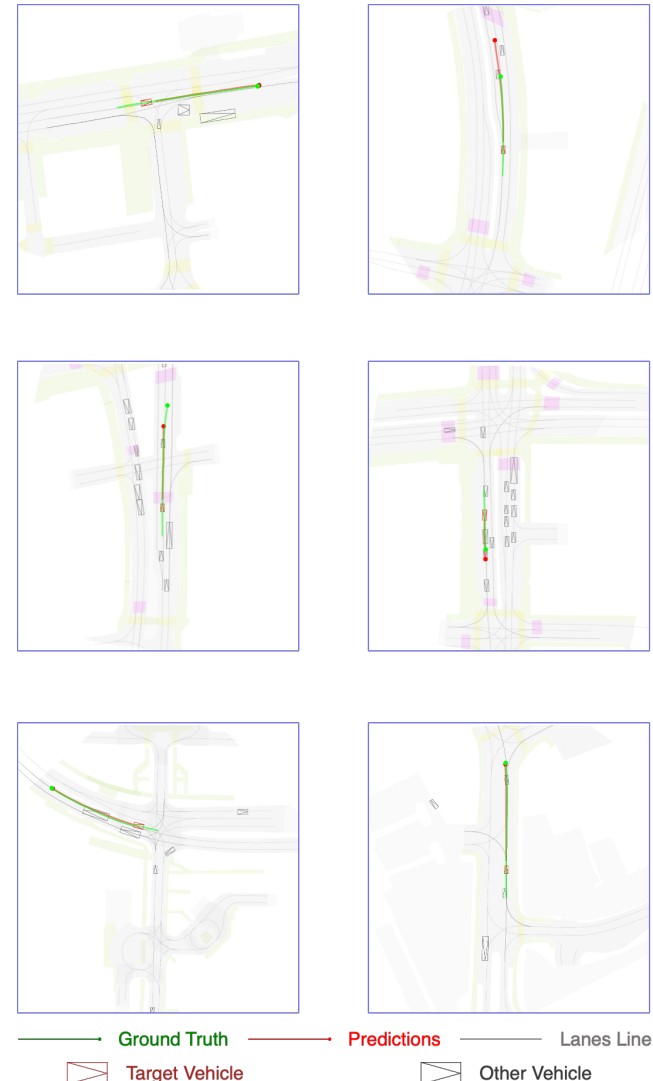

- $k = 1$

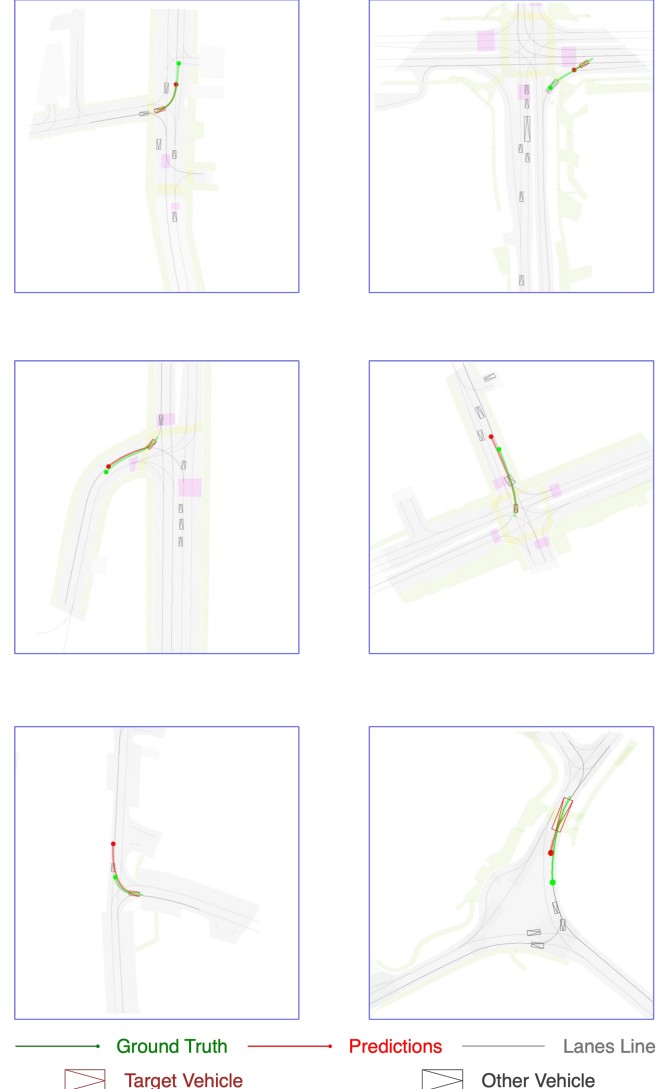

Ground Truth ———→  Predictions ———→  Lanes Line ———

Target Vehicle  Other Vehicle

- $k = 1$

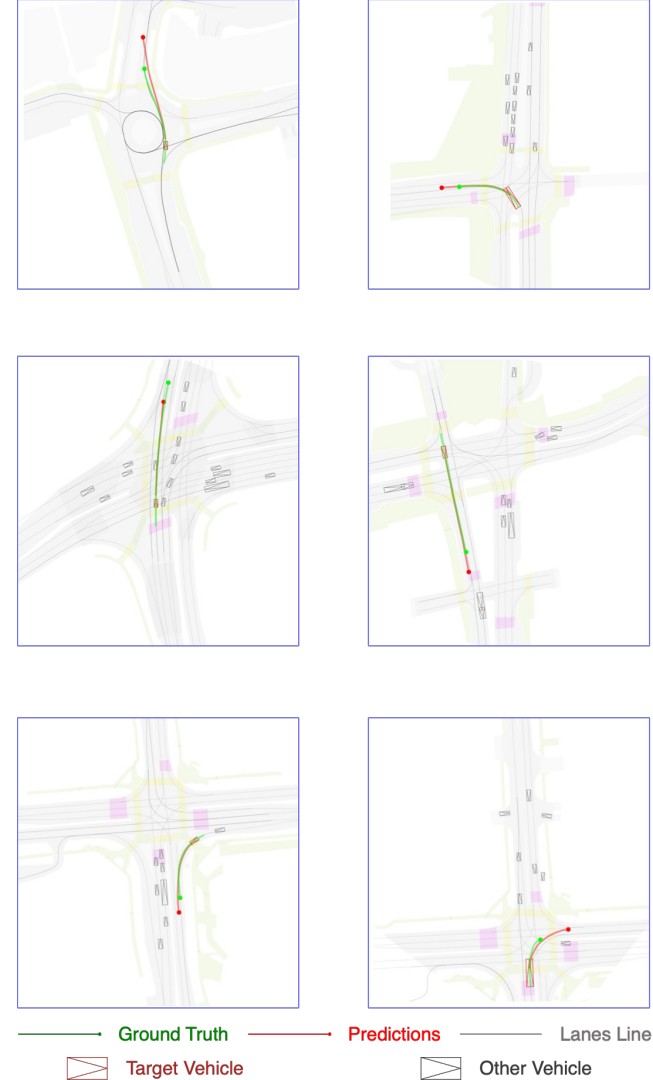

- $k = 1$

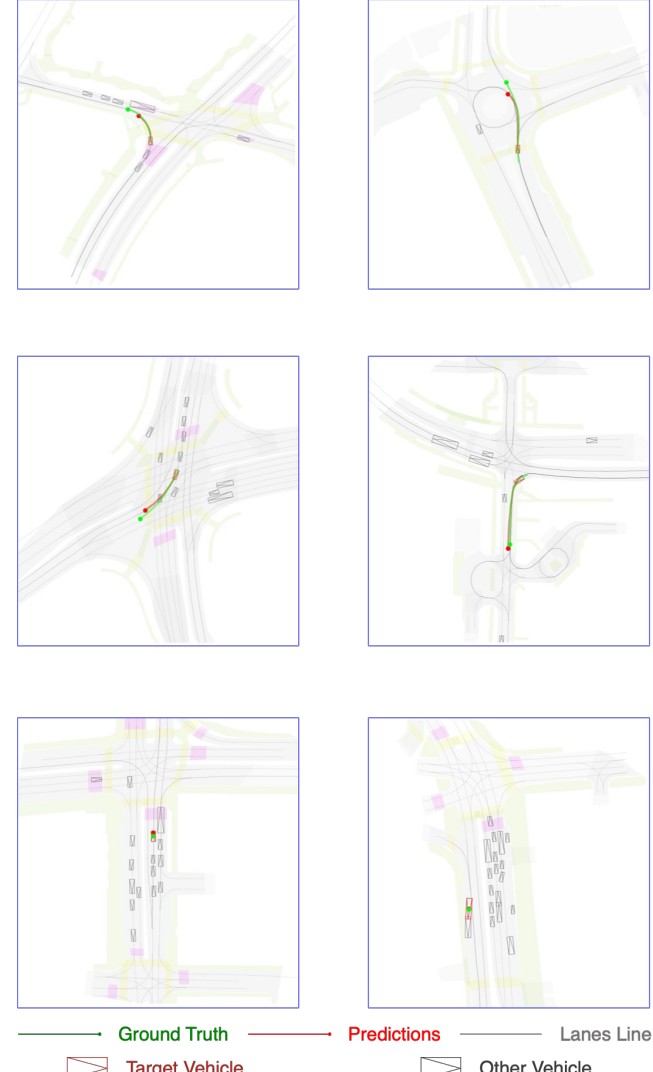

- $k = 5$

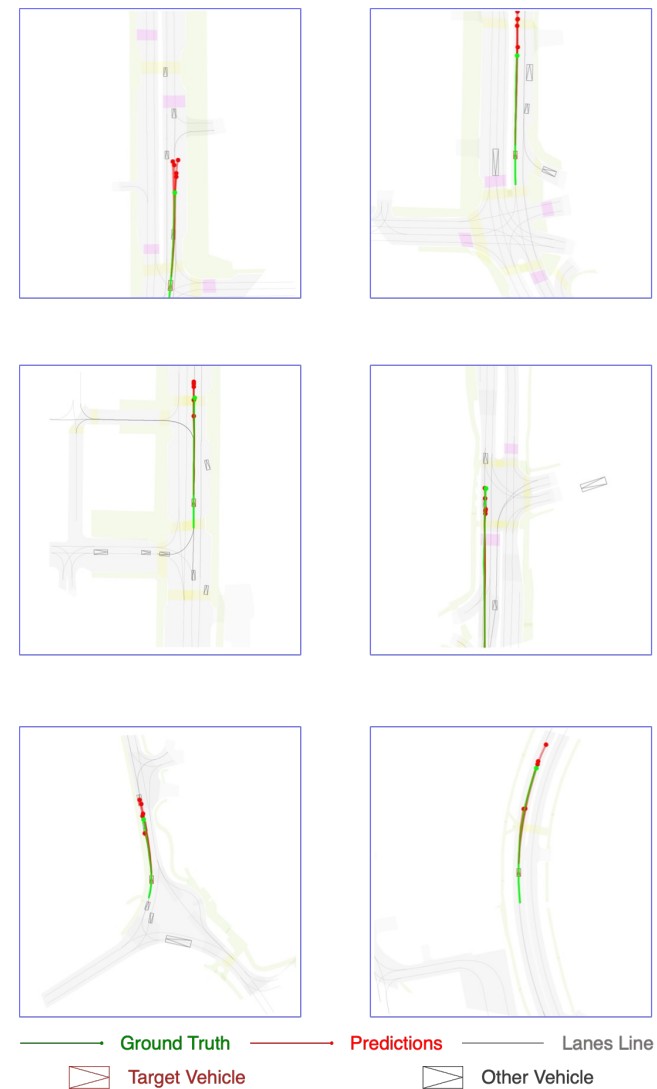

- $k = 5$

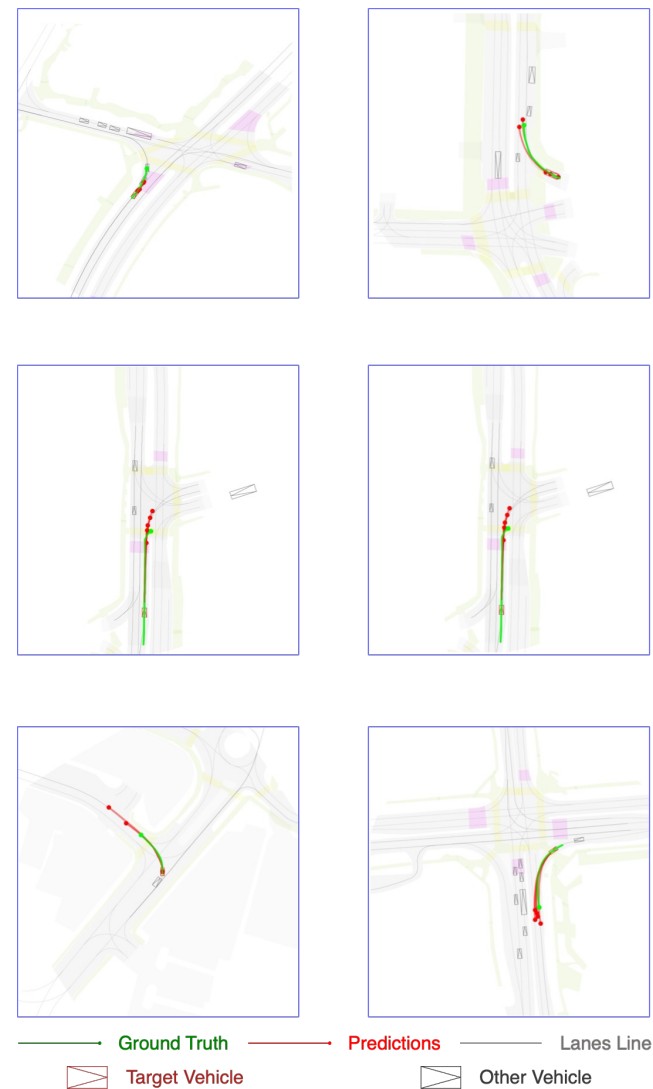

- $k = 5$

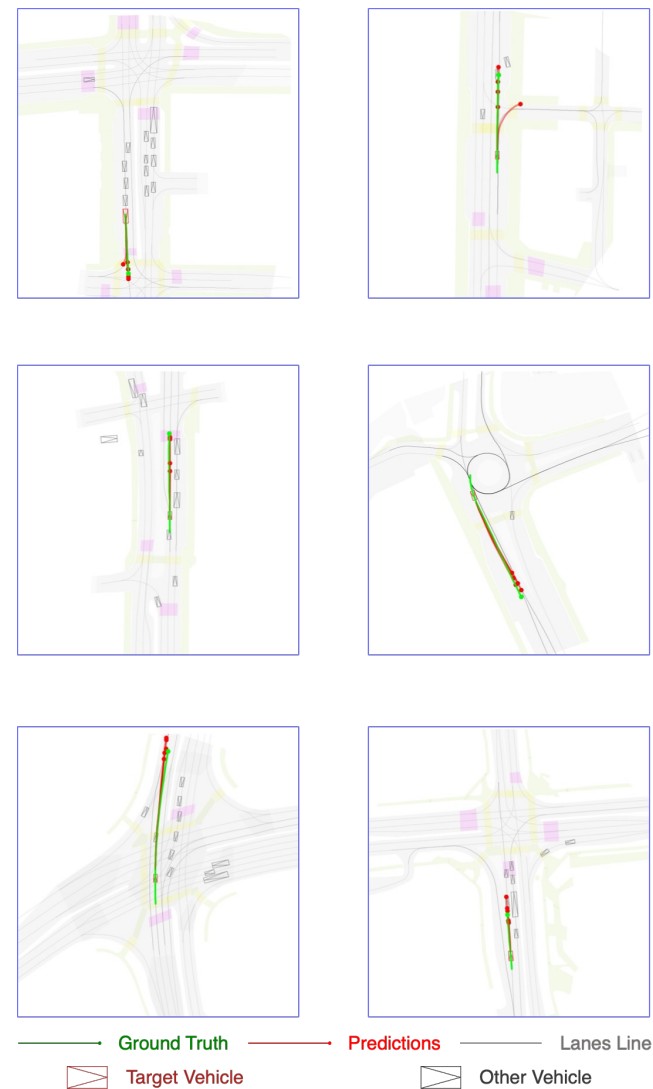

• $k = 5$

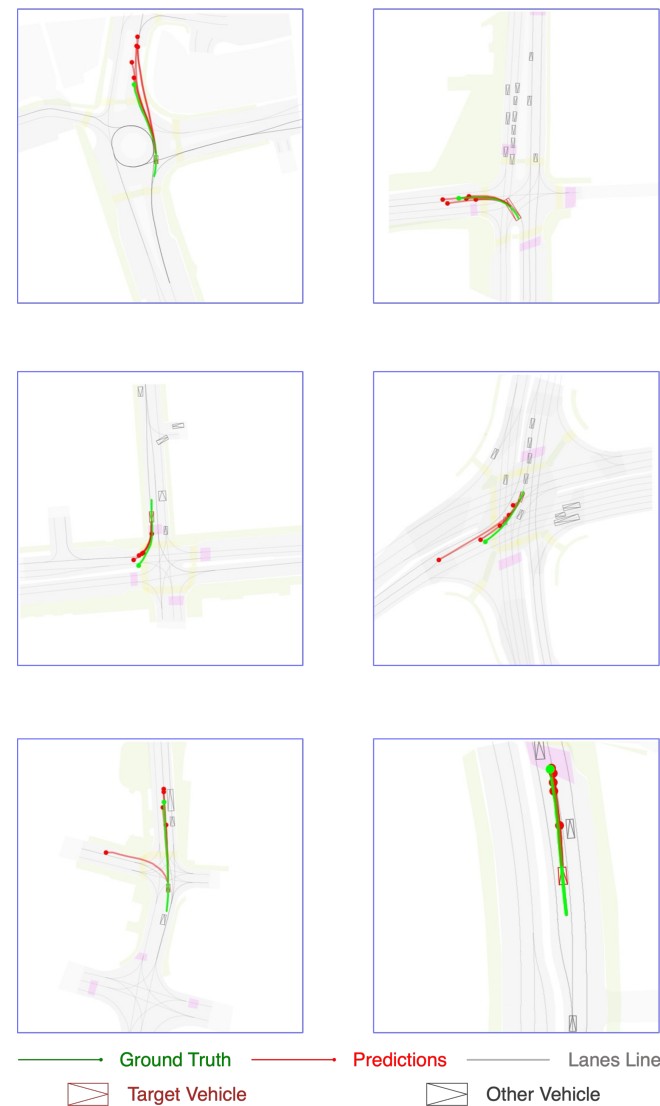

Ground Truth · Predictions · Lanes Line

Target Vehicle · Other Vehicle

- $k = 10$

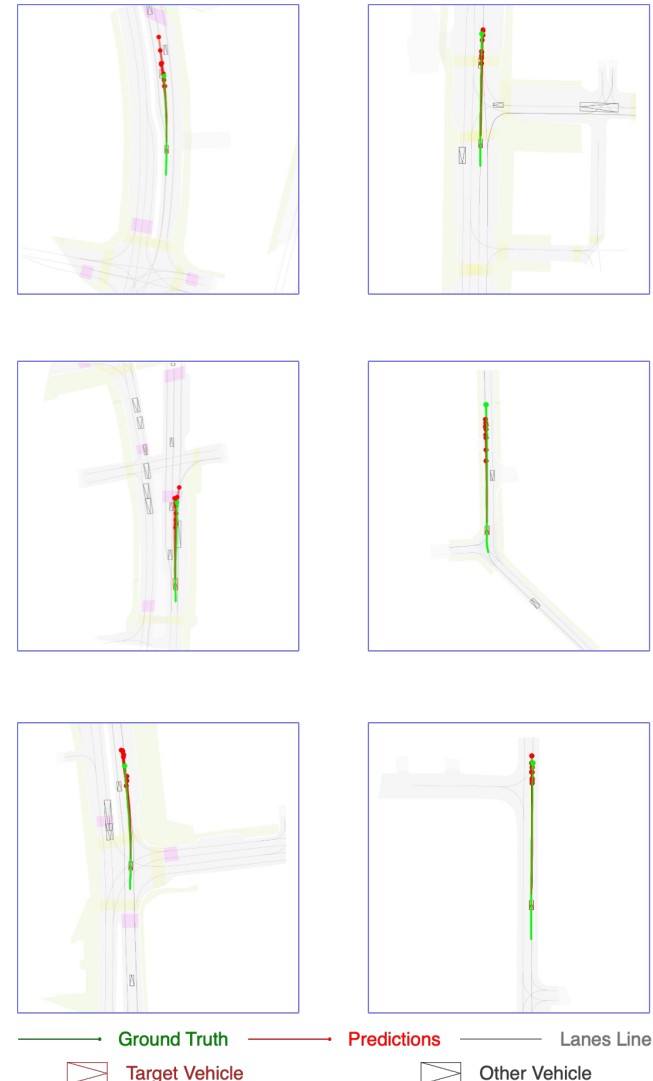

• $k = 10$

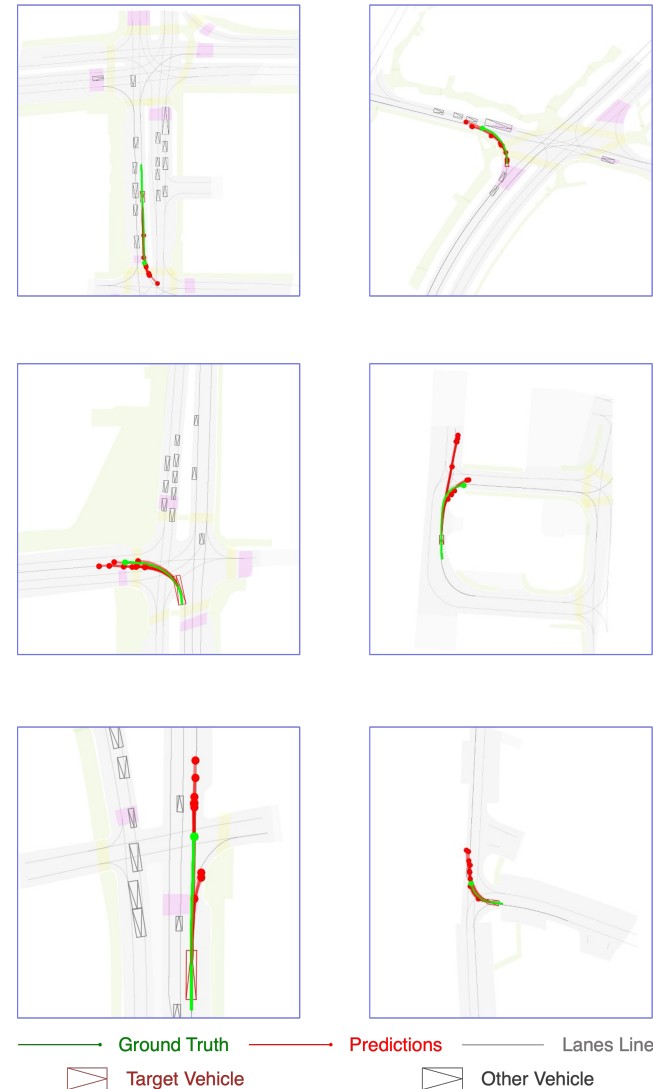

- $k = 10$

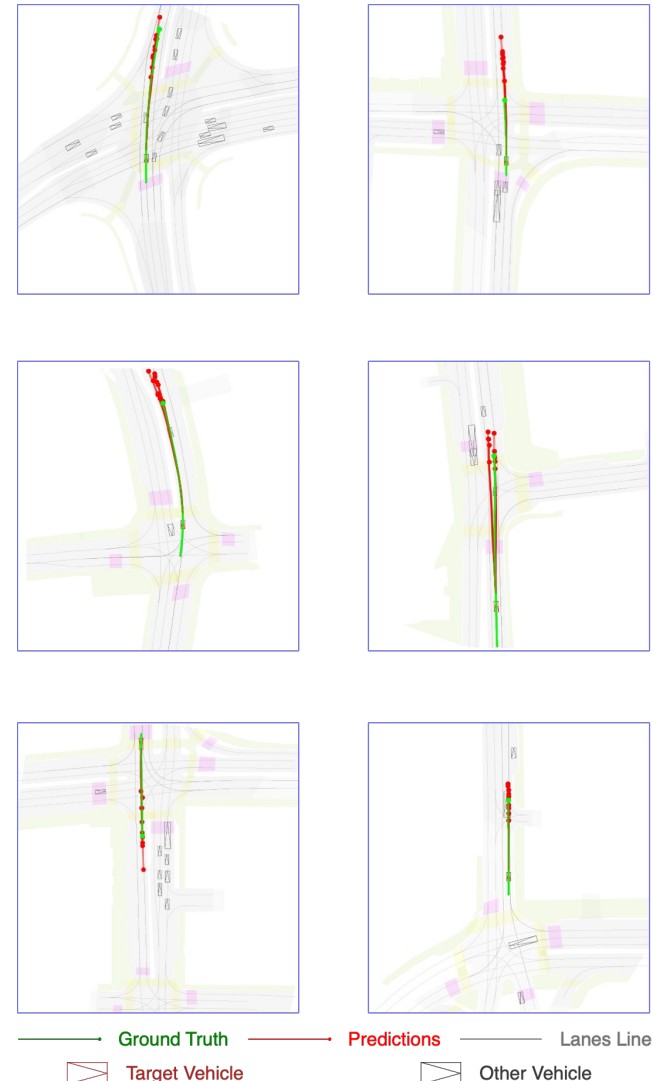

- $k = 10$

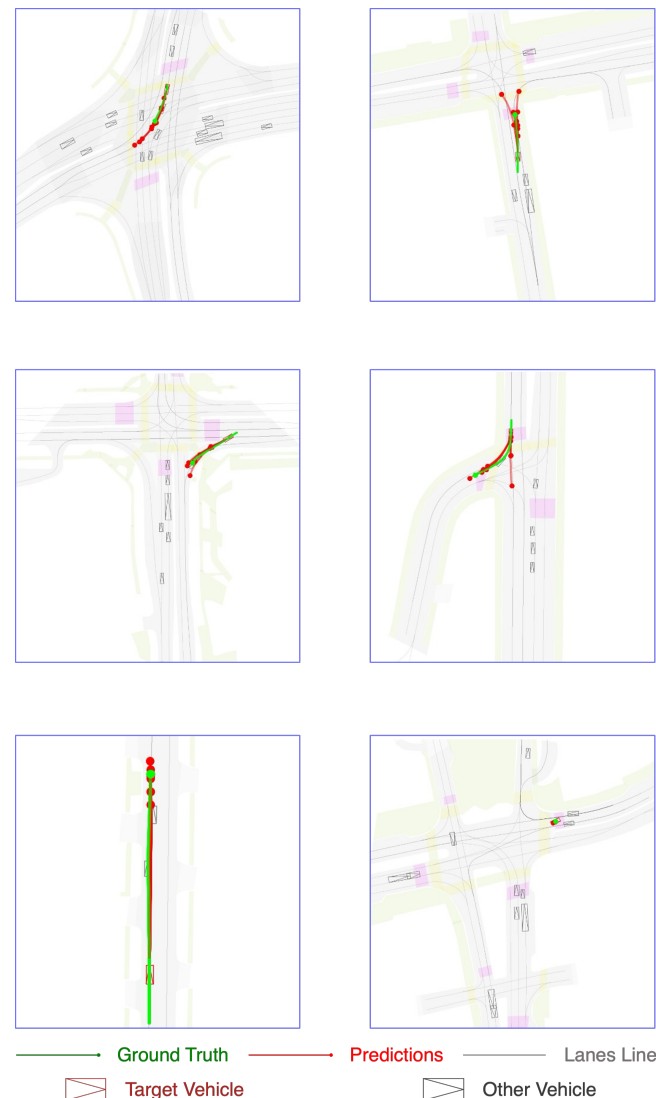

Ground Truth     Predictions     Lanes Line

Target Vehicle     Other Vehicle

