# OpenReview forum: "A Driving-Style-Adaptive Framework  for Vehicle Trajectory Prediction"
_NeurIPS.cc/2025/Conference — NeurIPS 2025 poster_

### Official Review · Reviewer_3sDS · 2025-06-27

**Clarity:** 3
**Significance:** 3
**Originality:** 3
**Rating:** 5
**Confidence:** 4

**Summary:**

The paper proposes an adaptive framework for predicting trajectories of vehicles of interest by incorporating their driving behavior into trajectory prediction. Incorporating driving behavior is achieved by pre-selected basis functions for the KAN-based method. These basis functions capture the driving styles of conservative, aggressive and normal drivers and are polynomially combined by the proposed method to capture “mixed” driving styles. Extensive comparisons with the SOTA and a comprehensive ablation study demonstrate the method's capability for driving-behavior-based prediction of trajectories

**Questions:**

S1: Add error bars to your results to demonstrate the stability of the method. This will further improve the quality of the evaluation presented.

S2: Integrate a summary of the limitations into the main text for a more conclusive understanding of the presented work.

S3: Figure 2 is crucial for understanding the method. To increase clarity and ensure that the flow from input to output is immediately apparent, the authors may consider improving Figure 3 (e.g., numbering the individual components or reorganizing the arrows, or some better solution) to illustrate the sequence of operations within the method in an improved manner.

Q1: The proposed method seems to use historical trajectories to incorporate the driving situation to some extent. However, the driving style is strongly influenced by a broader context. Have the authors considered incorporate the current situation in a more comprehensive manner?

Q2: The authors point out in the appendix (limitations section) that more driving styles can be included: Have the authors considered scaling the proposed method by identify and incorporate driving styles in an automatic manner rather than predefining the driving styles?

**Ethical Concerns:**

["NO or VERY MINOR ethics concerns only"]

**Final Justification:**

Due to the thorough rebuttal, which significantly improved the work, and the detailed answers to the questions raised during the discussion, the reviewers' confidence in accepting the article increased during the discussion phase. Issues regarding error bars, the relocation of important limitations from the appendix to the main text, and minor issues regarding the overall clarity of the article were resolved during this phase. The error bars and the additional experiments to clarify the pre-defined driving style using clustering had the greatest impact on the reviewers' positive assessment of the rebuttal.

**Limitations:**

No, although the authors have discussed the limitations in detail in the appendix, in the reviewer's view the limitations should be discussed in the main text (e.g. in the “Summary” section). Including a summary of the limitations discussed in the appendix in the main text with the most important limitations would be appropriate.

**Paper Formatting Concerns:**

No concerns.

**Quality:**

3

**Strengths And Weaknesses:**

$\textbf{Quality:}$

$\textit{Strengths:}$ The submission is presented in a technically sound manner that is further extended in the appendix. Theoretical claims are supported by comprehensive experiments on relevant datasets (Waymo, nuScenes, Agroverse) and are compared to various baselines that capture the field sufficiently.

$\textit{Weaknesses:}$ Although the evaluation is comprehensive, it lags error bars on the presented results. The authors argue that prior work is followed by not incorporating error bars. However, this should not be an argument for not incorporating error bars to, e.g., assess the stability of the achieved results and the proposed components. The reviewer suggests adding error bars to improve the quality of evaluation further. The review hopes this is achievable in the remaining time but acknowledges the limits of the timeframe. Also, the authors should be a bit more careful about including the weaknesses/limitations of the method in the main text. Limitations are presented in detail in the appendix but should (in the eyes of the reviewer) be incorporated in the main text to present a conclusive picture of the proposed method.


$\textbf{Clarity:}$

$\textit{Strengths:}$ The writing of the paper is clear and good to understand. Although the paper is heavier on math, the reasoning behind the method can be followed throughout the entire text. The results are presented in well-organized tables and figures.

$\textit{Weaknesses:}$ Figure 2 could be improved to increase clarity: The flow from input to output within the DSA framework is not clear at first glance. Numbering the individual components or restructuring the arrows could be a way to improve the flow and to understand when which part of the method is performed.

Furthermore, there are small formatting errors that can be improved. Here are some examples that the reviewer found:

-	Line 76, Eq (2), Line 77, Line 83, Line 95, Figure 4: Remove the yellow color

-	Line 77: Additional space after the citation [39,40]

-	Figure 2 description: Missing character (ndicates)

-	Line 186:  two times “.”


$\textbf{Significance/Originality:}$

$\textit{Strengths:}$ Driving styles have a significant impact on ego vehicles and other traffic participants. Various studies investigate driving behavior for autonomous driving, often showing that the impact of driving behavior should not be neglected. Therefore, the reviewer thinks that incorporating driving behavior into deep learning methods is a solid way to further advance the field. The combination of KANs and basic functions to achieve this incorporation is well-articulated by the authors and it will be interesting to see how further work extends this approach.

$\textit{Weaknesses:}$ The driving styles of traffic participants depend on the current temporal situation. For the proposed method, this situation is incorporated in a limited way through the historical trajectory of the vehicles (as far as the reviewer understands). A more comprehensive inclusion of the current situation (e.g. through a visual component that encodes the current situation (surroundings, weather, etc.) may have a high impact on driving style prediction. The reviewer understands that this probably lies beyond the scope of the paper. Therefore, this could be discussed in the limitations / future work section.

---

> ### Author Rebuttal · Authors · 2025-07-30
>
> We are very grateful for the 3sDS's professional and meticulous comments. They have been instrumental in refining our presentation and strengthening the paper's scientific contributions.
>
> S1: Add error bars to your results to demonstrate the stability of the method.
>
> A: The evaluation metrics reported in Table 1 represent the mean ± standard deviation over five random seeds for the Argoverse (A) and nuScenes (N) datasets, and over three seeds for the Waymo (W) dataset.
>
> Table 1. Multi-seed evaluation results on Argoverse (A), nuScenes (N), and Waymo (W). Metrics are reported as mean ± standard deviation.
>
> |     Dataset    |   ADE$_1$    |   FDE$_1$    |   ADE$_5$    |   FDE$_5$    |  ADE$_{10}$   |  FDE$_{10}$   |
> |:-------------:|:---------:|:---------:|:---------:|:---------:|:--------:|:--------:|
> |       N       | 2.69±0.08 | 6.47±0.15 | 1.21±0.07 | 2.74±0.03 | 0.85±0.02 | 2.00±0.01 |
> |    **Dataset**    |   **A-ADE**$_1$    |   **FDE**$_1$    |   **ADE**$_6$    |   **FDE**$_6$    | **W-minADE**   | **W-minFDE**   |
> |     A & W     | 1.17±0.05 | 2.85±0.11 | 0.62±0.02 | 0.95±0.01 | 0.5852±0.0152 | 1.1431±0.0227 |
>
> S2: Integrate a summary of the limitations into the main text for a more conclusive understanding of the presented work.
>
> (2) We will add the limitations in the main paper in the final version as follows:
>
> **Limitation**  We evaluate our method on prediction horizons up to 9 seconds (1 second of history and 8 seconds of future), which is the longest duration available in current open-access vehicle trajectory datasets. However, for significantly longer horizons (e.g., over one minute), direct long-term prediction may be unreliable and would likely require segment-wise modeling or hierarchical strategies.
> In addition, external factors such as strong conditions (e.g., traffic signals and regulatory constraints) and soft conditions (e.g., weather, which is often unlabeled in current datasets) can also affect trajectory prediction. Incorporating these contextual cues remains an important direction for future work.
>
> S3: the authors may consider improving Figure 2 to illustrate the sequence of operations within the method in an improved manner.
>
> A: We will revise Figure 2 by incorporating arrows to enhance clarity. On the left side, arrows will connect the highest-weighted experts in the histogram to their corresponding driving styles. On the right side, arrows will indicate the association between the predicted degrees and their respective polynomial bases.
>
> Q1 & Significance/Originally Weaknesses: Have the authors considered incorporate the current situation in a more comprehensive (the current situation e.g.surroundings, weather, etc.) manner?
>
> A: Driving styles reflect the behavioral characteristics of a driver [1-4], which typically remain stable over short temporal spans. Existing vehicle trajectory prediction datasets record trajectories of less than 10 seconds (Table 2).
> However, as you pointed out, how driving styles evolve in more comprehensive or long-term scenarios is indeed a compelling and valuable question. To address this issue, we plan to extend our work in future research as follows:
>
> ## Trajectory slicing
> For long-term vehicle trajectory prediction tasks (e.g., over 1 minute), we propose to segment the full trajectory into multiple shorter slices. Our DSA framework will be applied to each slice individually, and the final trajectory will be reconstructed by composing these predicted slices sequentially.
>
> ## Fine-grained learning
> The trajectory slices in above can be segmented with either equal or variable lengths. For segments exhibiting similar behavioral patterns, we may merge them into longer slices to reduce redundancy. Conversely, irregular or highly dynamic segments can be treated with greater attention, enabling fine-grained feature learning and more accurate predictions in critical scenarios.
>
> ## Surrounding environment
> Although existing datasets do not fully align contextual information (e.g., other traffic participants, weather conditions) with vehicle trajectories, we can incorporate available partial contexts and treat them as impact conditions for future feature learning. With the development of more comprehensive datasets, these contextual factors can be leveraged to enhance the accuracy and generalization capabilities of our framework.
>
> Table 2: Comparison of existing vehicle trajectory datasets.
> "His" and "Pre" denote the durations of the historical and predicted trajectories, respectively, while "Total" refers to the overall recorded duration per vehicle.
>
> | Datasets         | Collect Locations                | His | Pre | Total |
> |:----------------:|:--------------------------------:|:---:|:---:|:-----:|
> | KITTI [5]        | Karlsruhe                        |  2  |  4  |   6   |
> | Apolloscapes [6] | Beijing, ShangHai and SHenZhen   |  3  |  3  |   6   |
> | nuScenes [7]     | Boston and Singapore             |  2  |  6  |   8   |
> | Argoverse [8]    | Miami and Pittsburgh             |  2  |  3  |   5   |
> | INTERACTION [9]  | China, Germany and Bulgaria      |  1  |  3  |   4   |
> | InD [10]         | German                           | 3.2 | 4.8 |   8   |
> | RounD [11]       | Aachen                           |  2  |  4  |   6   |
> | HighD [12]       | German                           | 2.8 | 2.8 |  5.6  |
> | Waymo [13]       | USA                              |  1  |  8  |   9   |
>
> [1] Yao X, et al. Driving heterogeneity identification using machine learning: A review and framework for analysis[J]. TRIP, 2025.
>
> [2] Zheng X, Yang P, et al. Real-time driving style classification based on short-term observations[J]. IET Communications, 2022.
>
> [3] Garrosa M, et al. Holistic vehicle instrumentation for assessing driver driving styles[J]. Sensors, 2021.
>
> [4] Xu L, Hu J, et al. Establishing style-oriented driver models by imitating human driving behaviors[J]. IEEE TITS, 2015.
>
> [5] https://www.cvlibs.net/datasets/kitti/.
>
> [6] https://apolloscape.auto/.
>
> [7] https://www.nuscenes.org/nuscenes.
>
> [8] https://www.argoverse.org/.
>
> [9] https://interaction-dataset.com/.
>
> [10] https://levelxdata.com/ind-dataset/.
>
> [11] https://levelxdata.com/round-dataset/.
>
> [12] https://levelxdata.com/highd-dataset/.
>
> [13] https://waymo.com/open/.
>
> Q2: Have the authors considered scaling the proposed method by identify and incorporate driving styles in an automatic manner rather than predefining the driving styles?
>
> A: We consider adopting (1) semi-supervised and (2) unsupervised method for automatic categorization, and (3) evaluating predefined categories under multiple values beyond our proposed three-category setting.
>
> (1) **Semi-supervised approach**. Based on a set of partially predefined driving styles, we define a similarity threshold $c_{\text{style}}$ for each style. Trajectories with similarity scores below the corresponding $c_{\text{style}}$ are assigned to that predefined style. In contrast, those exceeding the threshold are categorized as newly discovered or blended styles, enabling further refinement and enrichment of the driving style taxonomy.
>
> (2) **Unsupervised approach**. We extract trajectory features such as speed, acceleration and other motion attributes. Then, we employ deep clustering-based methods, such as SharPer [14] and SMYRF [15] to automatically partition the driving styles into distinct categories.
> The optimal number of categories is determined using standard clustering evaluation metrics [16].
>
>
> (3) We evaluate multiple $k$-category settings using K-means clustering, and report the corresponding metrics (log-normalized except for Silhouette) in Table 3.
>
> Table 3. Clustering evaluation results on Argoverse (A) and nuScenes (N). A value of “1” indicates the best performance among all settings.
>
> | $k$ | A-$\Delta$WSS | Silhouette | $\Delta$CHI | N-$\Delta$WSS | Silhouette | $\Delta$CHI |
> |:---:|:-------------:|:----------:|:-----------:|:-------------:|:----------:|:-----------:|
> |  2  |               | 0.919      |             |               | 0          |             |
> |  3  | 1             | 1          | 0.810       | 1             | 0.700      | 1           |
> |  4  | 0.743         | 0.986      | 1           | 0.586         | 1          | 0.387       |
> |  5  | 0.022         | 0.283      | 0           | 0.287         | 0.268      | 0.324       |
> |  6  | 0             | 0          | 0.601       | 0             | 0.398      | 0           |
>
> Our three-category configuration yields the highest number of best scores across the metrics, supporting the validity and rationality of our chosen driving style taxonomy.
>
> Note: For all metrics listed below, larger values indicate better clustering performance.
>
> Note: For all metrics listed below, larger values indicate better clustering performance.
>
> -$\Delta$WSS (Within-Cluster Sum of Squares): Measures the improvement in intra-cluster compactness. A higher value suggests tighter grouping of samples within clusters after clustering.
>
> -Silhouette: Reflects both the cohesion within a cluster and the separation between clusters. A higher silhouette score indicates that samples are well matched to their own cluster and poorly matched to neighboring clusters.
>
> -$\Delta$CHI (Calinski–Harabasz Index): Captures the variation in inter-cluster separability and intra-cluster compactness. Higher values indicate better-defined and more distinct clusters.
>
> [14] Amiri M J, Agrawal D, El Abbadi A. Sharper: Sharding permissioned blockchains over network clusters[C]//sigmod. 2021.
>
> [15] Daras G, Kitaev N, Odena A, et al. Smyrf-efficient attention using asymmetric clustering[J]. NIPS, 2020.
>
> [16] Vysala A, Gomes J. Evaluating and validating cluster results[J]. arXiv, 2020.
>
> Small Formatting Corrections:
>
> -Remove the highlight on $\psi \left( x \right)$, the extra comma (Line 186), and the extra space (Line 77).
>
> -Figure 2: Correct to …while the arrow "indicates" the direction (history or future)…

---

> > ### Comment · Reviewer_3sDS · 2025-08-01
> >
> > The reviewer thanks the authors for addressing the suggestions and questions so thoroughly. The effort put into the revision is greatly appreciated, and the reviewer feels that the concerns have been adequately addressed. It is important to keep in mind the limitations of the current datasets regarding the time horizon and driving style for future work.
> >
> > Additionally, it was interesting to see the clustering-based results presented in the revision. Could the authors comment on whether the three clusters identified in the best configuration are similar to the chosen predefined base driving styles of the method?
> >
> > Regarding the incorporation of situational information into driving style-based predictions, the reviewer would like to discuss how this information could enhance performance over short temporal spans. For example, in rainy conditions, drivers may exhibit more cautious behavior, such as reducing speed and increasing following distance. The model may benefit from this additional context to better infer the driving style during brief intervals, as it may be more likely that vehicles will drive passively in such situations.

---

> > > ### Author Response · Authors · 2025-08-04
> > >
> > > Thank you for 3sDS's interest and thoughtful discussion of our work. We sincerely appreciate your valuable feedback. We will further enhance our model to better handle longer trajectories and more complex scenarios.
> > >
> > > 1. Whether the three clusters identified are similar to predefined in method?
> > >
> > > A: We illustrate these similarities through two approaches: (1) external comparison and (2) internal comparison.
> > >
> > > (1) We compute the cosine similarity between the trajectory associated with the highest-weighted polynomial and the predefined driving style standards [1–4], which include Conservative (Con), Aggressive (Agg), and Normal (Nor) drivers.
> > >
> > > Table 1. Cosine similarity reflecting consistent matching between polynomial based predictions and driving style standards on the Argoverse dataset.
> > > | $p_n$  | Con | Agg | Nor |
> > > |:--------:|:-------:|:-------:|:-------:|
> > > | $B_n$ | 0.884   | 0.129   | 0.391   |
> > > | $T_n$ | 0.167   | 0.927   | 0.329   |
> > > | $L_n$ | 0.206   | 0.333   | 0.963   |
> > >
> > > [1] https://www.sciencedirect.com/science/article/pii/S2590198225001903.
> > >
> > > [2] https://ietresearch.onlinelibrary.wiley.com/doi/pdfdirect/10.1049/cmu2.12405.
> > >
> > > [3] https://www.mdpi.com/1424-8220/21/4/1427.
> > >
> > > [4] https://ieeexplore.ieee.org/abstract/document/7078931/.
> > >
> > > (2) We compute the correct matching rate, defined as the proportion of cases in which the top‑weighted expert (Top‑1) aligns with the predefined polynomial for each driving style. The results appear in Table 2 and involve the polynomials Bernstein $B_n$, Chebyshev $T_n$ and Legendre $L_n$.
> > >
> > > Table 2. Matching accuracy between driving style categories and their predefined polynomial bases. Each percentage represents the fraction of cases in which the corresponding polynomial $p_n$ receives the highest expert weight within its respective driving style.
> > > | style | Argoverse | nuScenes | Top-1 |
> > > |:-----:|:---------:|:--------:|:-----:|
> > > | Con   | 86.15%    | 87.96%   | $B_n$    |
> > > | Agg   | 92.87%    | 94.26%   | $T_n$    |
> > > | Nor   | 80.04%    | 83.56%   | $L_n$    |
> > >
> > >
> > > 2. The model may benefit from this additional context to better infer the driving style during brief intervals.
> > >
> > > A: We outline potential integration schemes and their expected outcomes by incorporating additional contextual information into the existing blocks of our framework.
> > >
> > > ## Modulation Factor
> > > The modulation factor quantifies how vehicle motion dynamics change under different context conditions. Weather exemplifies this dependency. Drivers vary their behavioral magnitude depending on weather context [5]. Studies show that road friction coefficients take values of 0.5 on wet, 0.4 in rain, 0.28 in snow, and 0.18 on icy surfaces, respectively [6].
> > > We compute the corresponding trajectory changes in speed, acceleration for each driving style based on the friction modulation values.
> > >
> > > ## Multi-Modal Feature Fusion
> > > We treat contextual information as an instantaneous driving style modulator. By encoding additional contextual cues and integrating them into our framework, we assess which types of context exert the most influence on driving style prediction. For instance, the presence of brake lights from the leading vehicle may trigger a sudden shift toward conservative behavior due to a driver’s reactive attention [7].
> > > Additionally, the motion of surrounding agents such as pedestrians or cyclists, can dynamically alter driving styles. To capture this, we utilize a two-stream parallel prediction architecture [8], where one stream models the traffic scene and the other predicts trajectory, enabling the framework to account for rapid contextual changes and their impact on driving behavior.
> > >
> > > ## Context-Aware Gating Network
> > > We can design a context-aware gating network to dynamically reweight the expert (driving styles) outputs. The contextual information is incorporated into the gating network through various fusion strategies, such as concatenation, an independent branch or attention-based mechanisms.
> > >
> > > For instance, in traffic congestion scenarios, the context can be encoded as indicator vectors, where higher values denote more severe congestion. Under such conditions, drivers are more prone to exhibit extreme behaviors, shifting toward a more aggressive style [9]. This is reflected in the expert weight transitioning from (Con, Agg, Nor) $= (0.22, 0.50, 0.28)$ to $(0.13, 0.68, 0.19)$.
> > >
> > > Furthermore, prior work suggests that the “aggressive coefficient” grows with waiting time $\tau$ following a power-law relationship ($A \equiv N \tau^{\sigma}$, [9]) indicating increased aggression under prolonged delay [10].
> > >
> > > [5] https://ieeexplore.ieee.org/stamp/stamp.jsp?arnumber=10870573.
> > >
> > > [6] https://pmc.ncbi.nlm.nih.gov/articles/PMC11861737/.
> > >
> > > [7] https://openreview.net/forum?id=LLWj8on4Rv.
> > >
> > > [8] https://proceedings.neurips.cc/paper_files/paper/2024/file/8ec078518dcce6be1324cfd3de11ed24-Paper-Conference.pdf.
> > >
> > > [9] https://scholarship.rollins.edu/cgi/viewcontent.cgi?article=1046&context=as_facpub.
> > >
> > > [10] https://arxiv.org/pdf/1804.10965.

---

> > > > ### Comment · Reviewer_3sDS · 2025-08-06
> > > >
> > > > The reviewer would like to express gratitude to the authors for their thorough and detailed feedback on the additional questions and for discussing potential future work. Especially, the similarities between the predefined styles and the styles found through clustering are very interesting to see. Furthermore, the situation-awareness based on dynamic reweighting seems intriguing, especially when viewed through a practical and modular standpoint. The reviewer is looking forward to work building upon this work.

---

> > > > > ### Author Response · Authors · 2025-08-06
> > > > >
> > > > > We are truly grateful for your in-depth and constructive insights. Your observations, particularly on style clustering similarities and the potential of situation-aware dynamic reweighting have prompted deeper reflection and point to a promising direction for future work.

---

### Official Review · Reviewer_5qot · 2025-07-03

**Clarity:** 3
**Significance:** 2
**Originality:** 3
**Rating:** 4
**Confidence:** 4

**Summary:**

This paper proposes a Driving-Style-Adaptive (DSA) Framework by leveraging Kernel-based Adaptive Networks (KANs) with polynomial basis operators tailored to different driving styles. Bernstein, Chebyshev, and Legendre polynomials are used to model conservative, aggressive, and normal driving behaviors, respectively. A Mixture-of-Experts (MoE-TopK) strategy enables dynamic combination of these styles, reflecting real-world behavioral complexity. SMAC3 is used for hyperparameter optimization, particularly to choose effective polynomial degrees. Experiments on nuScenes, Argoverse, and Waymo show notable improvements over baselines.

**Questions:**

1. How are multi-modal predictions generated during experiments? Are multiple hypotheses sampled, or is a single deterministic output used for evaluation?
2. Is the polynomial degree fixed or adaptively updated during training? How is the chosen degree used during inference?
3. Additional experiments would strengthen the paper. For instance, it is not demonstrated whether specific polynomial operators (e.g., Bernstein, Chebyshev, Legendre) consistently correspond to particular driving styles in practice. An interpretability analysis that shows the expert contributions in different scenarios—especially those exhibiting distinct styles—would help validate the core claim. Furthermore, it would be valuable to explore whether increasing the diversity of basis operators (e.g., adding more types of polynomials) leads to better performance or simply increases model complexity without clear gains.

**Ethical Concerns:**

["NO or VERY MINOR ethics concerns only"]

**Final Justification:**

The rebuttal answered most of my questions.

**Limitations:**

yes

**Quality:**

3

**Strengths And Weaknesses:**

Strengths:
1. Integrate KANs into vehicle trajectory prediction with explicit modeling of driving styles.
2. The use of distinct polynomial bases is conceptually linked to characteristics of driving behaviors (e.g., Bernstein’s smoothness for conservative driving).
3. MoE-TopK allows dynamic combination of styles, enhancing the model’s ability to handle diverse, multimodal behaviors.
4. The experiments are extensive. The framework is validated on three datasets and includes qualitative and ablation studies.

Weaknesses:
1. The model's interpretability is limited, as the contribution of each polynomial expert is not visualized or analyzed across scenarios, making it unclear if driving styles are explicitly learned or used during inference.
2. Despite the focus on modeling driving styles, the evaluation does not include scenarios that clearly reflect variations in driver behavior, missing an opportunity to demonstrate the model’s key advantage.
3. The justification for using Legendre polynomials to represent "normal" driving style is vague and overlaps with the conservative case, weakening the conceptual distinction between the styles.
4. The paper omits discussion of training and inference time, which is important given the added complexity of the mixture-of-experts architecture and degree selection procedure.

---

> ### Author Rebuttal · Authors · 2025-07-30
>
> We sincerely appreciate the 5qot's insightful comprehensively consider multiple aspects of our vehicle trajectory work.
>
> Question1: How are multi-modal predictions generated during experiments? Are multiple hypotheses sampled, or is a single deterministic output used for evaluation?
>
> A: Generating multi-modal predictions is equivalent to producing $k$ distinct trajectory hypotheses. This is reflected by the subscript $k$ in the evaluation metrics. By independently repeating the stochastic generation process $k$ times, the model outputs $k$ diverse trajectories, from which the best prediction is selected based on the evaluation criterion.
>
> Q2: Is the polynomial degree fixed or adaptively updated during training? How is the chosen degree used during inference?
>
> A: The degree is adaptive during training but fixed during inference. In the training phase, our framework learns the optimal degree for each expert (corresponding to a specific driving style). During inference, these learned degrees are used in combination to obtain appropriate expert weights.
>
> Q3 & Weaknesses 2: (1) Despite the focus on modeling driving styles, whether specific polynomial operators consistently correspond to particular driving styles in practice, missing an opportunity to demonstrate the model's key advantage.
>
> A: We calculate the cosine similarity between the trajectory corresponding to the highest-weighted polynomial $p_n$ and the predefined driving style standards [1-4], which include Conservative (Con), Aggressive (Agg) and Normal (Nor) categories. The results on the nuScenes dataset are presented below:
>
> Table 1. Cosine similarity reflecting consistent matching between polynomial based predictions and driving style standards on the nuScenes dataset.
> | $p_n$   |   Con   |   Agg   |   Nor   |
> |:--------:|:-------:|:-------:|:-------:|
> | $B_n$   |  0.953  |  0.158  |  0.489  |
> | $T_n$   |  0.266  |  0.987  |  0.353  |
> | $L_n$   |  0.310  |  0.263  |  0.992  |
>
> [1] Yao X, Calvert S C, Hoogendoorn S P. Driving heterogeneity identification using machine learning: A review and framework for analysis[J]. Transportation Research Interdisciplinary Perspectives, 2025.
>
> [2] Zheng X, Yang P, Duan D, et al. Real-time driving style classification based on short‐term observations[J]. IET Communications, 2022.
>
> [3] Garrosa M, Olmeda E, Fuentes del Toro S, et al. Holistic vehicle instrumentation for assessing driver driving styles[J]. Sensors, 2021.
>
> [4] Xu L, Hu J, Jiang H, et al. Establishing style-oriented driver models by imitating human driving behaviors[J]. IEEE Transactions on Intelligent Transportation Systems, 2015.
>
> Q3: (2) Furthermore, it would be valuable to explore whether increasing the diversity of basis operators (e.g., adding more types of polynomials) leads to better performance or simply increases model complexity without clear gains.
>
> A: We incorporate additional KAN variants: TaylorKAN [5], JacobiKAN [6], FourierKAN [7] and WaveKAN [8] into our KANlayer named "multi". The performance of these basis function combinations under the TopK=3 MoE setting is evaluated on the nuScenes dataset. The corresponding results are summarized in Table 2.
>
> Table 2. Comparison of multiple basis operators with our proposed method on the nuScenes dataset under the MoE TopK=3 setting.
>
> | Method | ADE$_{1}$ | FDE$_{1}$ | ADE$_{5}$ | FDE$_{5}$ | ADE$_{10}$ | FDE$_{10}$ |
> |:------:|:---------:|:---------:|:---------:|:---------:|:----------:|:----------:|
> | multi  |   2.81    |   **6.44**    |   1.46    |   **2.72**    |   0.91     |   2.08     |
> | Ours   |   **2.69**    |   6.47    |   **1.21**    |   2.74    |   **0.85**     |   **2.00**     |
>
> From Table 2, although the values of FDE$_1$ and FDE$_5$ show only marginal increases (at the thousandth level), all other metrics exhibit noticeable performance degradation compared to our proposed method. Due to the larger pool of experts, some are selected less frequently during training resulting in undertraining. This imbalance further leads to overfitting in the gating network, ultimately compromising the generalization ability of the model.
>
> [5] https://github.com/Muyuzhierchengse/TaylorKAN.
>
> [6] https://github.com/SpaceLearner/JacobiKAN.
>
> [7] https://github.com/GistNoesis/FourierKAN.
>
> [8] https://github.com/cconut/WaveKANet.
>
> Weaknesses 1: The model's interpretability is limited, as the contribution of each polynomial expert is not visualized or analyzed across scenarios, making it unclear if driving styles are explicitly learned or used during inference.
>
> A: We compute the proportions of each polynomial expert p_n being selected as the Top-1 weight as shown in Table 3. Here, $p_n$ consists of the Bernstein $B_n$, Chebyshev $T_n$, and Legendre $L_n$ polynomials.
>
> Table 3. Top-1 selection proportions for each polynomial expert.
>
> | $p_n$  | Argoverse | nuScenes |
> |:------:|:---------:|:--------:|
> | $B_n$ | 45.47% | 39.22% |
> | $T_n$ | 25.52% | 20.57% |
> | $L_n$ | 29.01% | 40.21% |
>
> These results indicate that all experts are utilized, with none having a selection rate of zero.
>
> W3: The justification for using Legendre polynomials to represent "normal" driving style is vague and overlaps with the conservative case, weakening the conceptual distinction between the styles.
>
> A: We justify the use of Legendre polynomials $L_n$ to represent the Normal driving style from two perspectives: i) empirical evidence and 2) mathematical proposition analysis.
>
> ## Experimental Results
>
> We conduct ablation experiments by replacing the $L_n$, originally used for Normal (Nor) drivers with Hermite polynomials $H_n$, and replacing the Bernstein polynomials $B_n$ (for Conservative drivers) with Charlier polynomials $C_n$. The performance comparison is reported in Table 4 (Argoverse, A-Method) and Table 5 (nuScenes, N-Method), respectively.
>
> Table 4. Replacement results on the Argoverse dataset (**best** results in bold).
>
> |   A-Method    | ADE1 | FDE1 | ADE6 | FDE6 |
> |:----------:|:----:|:----:|:----:|:----:|
> | Cn for Con | 1.48 | 3.76 | 1.10 | 1.53 |
> | Hn for Nor | 1.62 | 2.87 | 0.71 | 1.02 |
> |   ours     | **1.17** | **2.85** | **0.62** | **0.96** |
>
> Table 5. Replacement results on the nuScenes dataset (**best** results in bold).
>
> |  N-Method   | ADE1 | FDE1 | ADE5 | FDE5 | ADE10 | FDE10 |
> |:----------:|:----:|:----:|:----:|:----:|:-----:|:-----:|
> | Cn for Con | 2.79 | 6.69 | 1.32 | 2.81 | **0.79**  | 2.08  |
> | Hn for Nor | 2.81 | 6.53 | 1.48 | 2.94 | 1.01  | 2.23  |
> |   **ours** | **2.69** | **6.47** | **1.21** | **2.74** | 0.85  | **2.00** |
>
> Across the two datasets, our original configuration achieves 9 out of 10 best results. The only exception is a marginal 0.06 gap in ADE$_{10}$ on the nuScenes dataset. These results strongly support the validity of the proposed matching between driving styles and polynomial basis. Specifically, using $L_n$ for Normal drivers and $B_n$ for Conservative drivers.
>
> ## Mathematical Proposition analysis
> **Conservative drivers** typically maintain the lowest average speed among all driving styles and exhibit minimal behavioral variation. Their trajectories are characterized by smoothness and stability, with few abrupt changes. The uniform convergence property of $B_n$ ensures that the approximation error diminishes evenly across the interval, making them well-suited for modeling such stable, predictable trajectories.
>
> In contrast, **normal drivers** exhibit more regular but moderate changes in behavior. Their trajectories are generally smooth with consistent fluctuations, which are neither overly abrupt nor completely static. The $L_n$ known for their orthogonality and optimal approximation under the $L^2$ norm, are particularly effective for capturing such patterns. Their ability to represent smooth variations with computational efficiency makes them a suitable choice for approximating the trajectories of Normal drivers.

---

> > ### Comment · Reviewer_5qot · 2025-08-07
> >
> > The rebuttal addressed most of my concerns, with the following remaining:
> >
> > 1. The inference time is still not mentioned. Additionally, in the response, it is noted that the generation process needs to be rerun *k* times. Are these runs executed in parallel or sequentially? If done sequentially, does this make the inference very slow? I feel this is a very important aspect, as real-time performance is a crucial concern in autonomous driving.
> >
> > 2. In response to Q3 and Weakness 2, you have generated a set of predefined driving style standards. Could you provide an explicit summary of how they are generated, rather than citing references?

---

> > > ### Author Response · Authors · 2025-08-09
> > >
> > > We sincerely appreciate your highly professional and insightful review, which has prompted us to deepen and clarify our paper further. Thank you for giving us the valuable opportunity to strengthen our paper through this revision.
> > >
> > > 1. (1) The inference time is still not mentioned. Additionally, in the response, it is noted that the generation process needs to be rerun $k$ times. (2) Are these runs executed in parallel or sequentially? If done sequentially, does this make the inference very slow? I feel this is a very important aspect, as real-time performance is a crucial concern in autonomous driving.
> > >
> > > A:(1) We report the inference time (minutes:seconds) for each dataset, where “Traj” denotes the number of generated trajectories. The table also specifies the GPU type and the exact NVIDIA model used, as shown in Table 1.
> > >
> > > Table 1. Inference time for each dataset.
> > > | Datasets   | No.-GPU  |  Traj  |   Times    |
> > > |:----------:|:--------:|:------:|:----------:|
> > > | Argoverse  | 1-4090   | 39472  | 46 min 22 s |
> > > | nuScenes   | 1-4091   |  3280  |  6 min 47 s |
> > > | Waymo      | 2-4090   |  1290  |  8 min 03 s |
> > >
> > > (2) We execute the process in parallel, conceptually repeating the generation process $k$ times to ensure diversity. However, all trajectories are produced within a single forward pass rather than through sequential reruns of the model.
> > >
> > > To achieve this, we sample different MoE gates by injecting independent noise into the gating logits multiple times, thereby obtaining diverse expert weight distributions. Instead of performing $k$ sequential inference runs, we batch these noise realizations and process them in parallel within a single forward pass, efficiently generating $k$ distinct trajectories.
> > >
> > >
> > >
> > > 2. In response to Q3 and Weakness 2, you have generated a set of predefined driving style standards. Could you provide an explicit summary of how they are generated, rather than citing references?
> > >
> > > A:  The style standards are computed as follows:
> > >
> > > - Velocity: The instantaneous velocity of the vehicle.
> > >
> > > - Acceleration: The instantaneous acceleration of the vehicle.
> > >
> > > - Space headway: The distance from the front center of a vehicle to the front center of the preceding vehicle.
> > >
> > > - Time headway: The time required for a vehicle, traveling at its current speed to reach the front center of the preceding vehicle.

---

> ### Comment · Area_Chair_cUmV · 2025-08-01
>
> Dear Reviewer 5qot,
>
> You mentioned key concerns on the interpretability, evaluation, etc. Please check the rebuttal and provide feedback. Thank you.
>
> Best,
>
> AC

---

### Official Review · Reviewer_Hx7U · 2025-07-03

**Clarity:** 3
**Significance:** 2
**Originality:** 3
**Rating:** 5
**Confidence:** 4

**Summary:**

This paper introduces a novel Driving-Style-Adaptive (DSA) framework for vehicle trajectory prediction that accounts for the variability in human driving behavior. Instead of assuming a fixed driving style, the model categorizes drivers into three types of conservative, aggressive, and normal. It then approximates their behaviors using distinct polynomial families: Bernstein polynomials for conservative, Chebyshev polynomials for aggressive, and Legendre polynomials for normal driving. A mixture of experts mechanism dynamically combines these polynomial experts. The proposed method is evaluated on three benchmark datasets (nuScenes, Argoverse, and Waymo) and demonstrates strong performance.

**Questions:**

1- For Table 5 in Appendix B.2, could the authors clarify the number of training samples corresponding to each motion type (e.g., turn right, turn left)

2- Are the reported results based on the average of multiple runs, or do they reflect a single best run?

**Ethical Concerns:**

["NO or VERY MINOR ethics concerns only"]

**Final Justification:**

My main concern was whether the proposed method is capable of distinguishing between different driving styles. The authors have addressed this (and other concerns) clearly and sufficiently in their response to my review. I find their clarifications satisfactory and, based on this, I am updating my score to accept.

**Limitations:**

Not discussed in the paper.

**Quality:**

3

**Strengths And Weaknesses:**

Weaknesses:

1- The paper contains several typos that should be corrected.

Line 66: "similarity".

Line 132: Should be "at the expense of safety."

Line 223: "follo"

Line 269:  Missing "for" before "more"

Line 283: "effective"

Line 288: "for evaluate"

2- The paper lacks a visual analysis showing whether the system accurately distinguishes between different driving styles and whether the expert weights appropriately reflect those behaviors.

3- The paper does not appear to discuss training time or provide key implementation details such as optimizer settings, batch size, or number of training epochs. Similarly, the evaluation metrics and dataset characteristics (e.g., size, preprocessing) are not clearly described in the main paper or appendix.

4- No significance tests are presented to assess whether the performance improvements are statistically meaningful. Including such tests would strengthen the empirical claims and support the reported gains more robustly.

Strengths:

Paper is well written, with good flow and clarity. The introduction section is laid out well, and experiments are explained clearly. The method is interesting and novel.

---

> ### Author Rebuttal · Authors · 2025-07-30
>
> We greatly appreciate the Hx7U's detailed and constructive comments. These would help us better position our work.
>
> Question 1: Clarify the number of training samples corresponding to each motion type in Table 5 Appendix B.2.
>
> A: We present the rate of each motion class in Table 1, where the motion definitions follow the *ClassifyTrack* class schema of the Waymo dataset*.
>
> Table 1: The b-minFDE values and motion occurrence rates on the nuScenes dataset. The **best** results are highlighted.
>
> | Method Type | Stationary | Straight | Straight right | Straight left | Right U-turn | Right-turn | Left U-turn | Left-turn |
> |:------------|:----------:|:--------:|:--------------:|:-------------:|:------------:|:----------:|:-----------:|:---------:|
> | MTR         | 2.15       | 2.58     | **4.85**        | 4.26          | 8.13         | 4.82       | **5.17**     | **4.85**   |
> | DSA         | **2.03**   | **2.48** | 4.96           | **4.17**      | **8.11**      | **4.77**    | 5.28         | 4.87      |
> | Rate        | 66.94%     | 12.76%   | 6.56%          | 6.25%         | 0.11%        | 3.49%      | 0.16%        | 3.73%     |
>
> The weighted average b-minFDE is 2.719 for MTR and 2.627 for our framework, indicating that our method improves prediction accuracy by a margin of 0.092.
>
> *Waymo: https://github.com/waymo-research/waymo-open-dataset.
>
> Q2: Are the reported results based on the average of multiple runs, or do they reflect a single best run?
>
> A2: The result are the average value under five (Argoverse, nuScenes) and three (Waymo) seeds. The whole results are shown in Table 2.
>
> Table 2. Multi seeds results in Argoverse (A), nuScenes (N) and Waymo (W). The metrics are listed by average ± standard.
> |     Dataset    |   ADE$_1$    |   FDE$_1$    |   ADE$_5$    |   FDE$_5$    |  ADE$_{10}$   |  FDE$_{10}$   |
> |:-------------:|:---------:|:---------:|:---------:|:---------:|:--------:|:--------:|
> |       N       | 2.69±0.08 | 6.47±0.15 | 1.21±0.07 | 2.74±0.03 | 0.85±0.02 | 2.00±0.01 |
> |    **Dataset**    |   **A-ADE**$_1$    |   **FDE**$_1$    |   **ADE**$_6$    |   **FDE**$_6$    | **W-minADE**   | **W-minFDE**   |
> |     A & W     | 1.17±0.05 | 2.85±0.11 | 0.62±0.02 | 0.95±0.01 | 0.5852±0.0152 | 1.1431±0.0227 |
>
> Weakness 1: Several typos.
>
> A: We will modify it in final version.
>
> W2: The paper lacks a visual analysis showing whether the system accurately (1) distinguishes between different driving styles and whether the (2) expert weights appropriately reflect those behaviors.
>
> A: (1) Based on the characteristics of the three driving styles [1], we compute statistical measures for the datasets (Table 3). Specifically, we report the mean and variance (var) for key motion indicators, including velocity ($v$), acceleration ($a$), space headway ($S_{h}$), and time headway ($T_{h}$).
>
> Table 3. Statistical characteristics of driving styles in the Argoverse (A-Metric) and nuScenes (N-Metric) datasets.
>
> | A-Metric       | Con-Mean | Var    | Agg-Mean | Var    | Nor-Mean | Var    |
> |:-------------:|:--------:|:------:|:--------:|:------:|:--------:|:------:|
> | $v$           | 7.071    | 9.710  | 8.753    | 14.799 | 8.268    | 12.691 |
> | $a$           | 0.010    | 9.095  | 0.067    | 12.710 | -0.022   | 17.481 |
> | $S_{h}$       | 8.600    | 51.961 | 12.827   | 106.715| 11.040   | 86.566 |
> | $T_{h}$       | 1.859    | 4.936  | 2.177    | 15.911 | 2.041    | 9.669  |
>
> Note:
> $S_h$: Spacing provides the distance between the front centre of a vehicle to the front center of the preceding vehicle.
>
> $T_h$: The time to travel from the front center of a vehicle to the front center of the preceding vehicle.
>
> [1] Zheng X, et al. Real‐time driving style classification based on short‐term observations[J]. IET Communications, 2022.
>
> (2) To reflect the relationship between expert weights and driver behaviors (styles), we compute the **correct matching rate**, defined as the proportion of cases where the highest expert weight (Top-1) corresponds to the expected polynomial $p_n$ under each driving style. The results are reported in Table 4, where the first letter denotes each driving style.
>
> Table 4. The matching relationship between different driving styles and their corresponding $p_n$. The percentage indicates the rate at which each $p_n$ has the highest weight within each driving style.
>
> | style | Argoverse | nuScenes | Top-1 |
> |:-----:|:---------:|:--------:|:-----:|
> | Con   | 86.15%    | 87.96%   | $B_n$    |
> | Agg   | 92.87%    | 94.26%   | $T_n$    |
> | Nor   | 80.04%    | 83.56%   | $L_n$    |
>
> W3: (1) Training time or provide key implementation details.
>
> A: (1) The average training time per epoch (in minutes:seconds) across the three datasets is reported in Table 3. The second column indicates the number of GPUs used and their specific model (NVIDIA).
>
> Table 3. Average training time per epoch for each dataset.
>
> | Datasets   | No.-GPU |    Times     |
> |:----------:|:-------:|:------------:|
> | Argoverse  |  1-4090 |   1min 20s    |
> | nuScenes   |  1-4091 |   6min 52s   |
> | Waymo      |  2-4090 |  56 min 45 s |
>
> Key implementation details: The model is implemented using PyTorch. We employ single-layer KANs with three expert networks (TopK, K=3), and polynomial degrees ranging from 1 to 10. The initial weights for the distance loss and MoE-$K$ loss are both set to 0.5. Further implementation details will be provided in the final version.
>
> W3: The (2) evaluation metrics and (3) dataset characteristics (e.g., size, preprocessing).
>
> A(2): We will include the mathematical definitions in final version. Specifically,
>
> -- Average / Final Displacement Error (ADE / FDE):
>
> ADE $=\frac{1}{T} \sum\nolimits^{T}_{t=1} \| \widetilde{p_{t}} -p_{t}\|_{L^{2}}$ , FDE $=\| \widetilde{p_{T}} -p_{T}\|_{L^{2}} .$
>
> Where $T$ represents the prediction length. For $\widetilde{p_{t}}$ and $p$ are the predicted and ground truth $(x,y)$ positions respectively.
>
> -- Missing Rate (MR): The ratio of whether the Euclidean distance between the $\widetilde{p_{t}}$ and $p$ at the FDE for all predictions is larger than 2m.
>
> -- Brier-minFDE ($b$-minFDE): $\text{bminFDE} = \text{minFDE} + (1 - p)^2$, where $p$ is the probability of probability of the best forecasting trajectory with minimum endpoint error (minFDE).
>
> (3) We preprocess the three datasets using the official pip packages provided by their respective baselines. The characteristics of each dataset are summarized in Table 4. The preprocessing procedures are described below.
>
> Table 4. Characteristics of the evaluation datasets.
>
> | Datasets      | Collect                    | Size    | Library                     | Select                 |
> |:-------------:|:--------------------------:|:-------:|:---------------------------:|:----------------------:|
> | Argoverse [2] | Miami and Pittsburgh       |    23.69G     | Argoverse API / devkit      | Motion Forecasting     |
> | nuScenes [3]  | Boston and Singapore       |    4.81G     | nuscenes-devkit             | Motional-Full dataset  |
> | Waymo [4]     | Phoenix, AZ, Kirkland etc. | 83.50G  | waymo-open-dataset [5]      | Motion1.1-scenario     |
>
> [2] https://www.argoverse.org/index.html.
>
> [3] https://www.nuscenes.org/nuscenes.
>
> [4] https://waymo.com/open/.
>
> [5] https://github.com/waymo-research/waymo-open-dataset.
>
> Preprocessing is based on the corresponding official libraries.
> First, extract trajectory features (position, velocity, acceleration) and corresponding map elements (e.g., lane lines, traffic lights).
> Then, align them by timestep to ensure data integrity and output.
>
> W4: Significance tests to assess whether the performance improvements are statistically meaningful.
>
> A: To verify the robustness of our improvements over the baseline, we conduct two analyses: (1) comparison against existing methods and (2) self-evaluation under multiple random seeds.
>
> (1) We perform $t$-tests against Context-Aware [6], the most recent baseline method with publicly available code. For fairness, we use the metrics reported in their original paper. The results are presented in Table 5.
>
> Table 5. Multi-seed results on the nuScenes dataset. "No." indicates the seed value. Columns 2–5 show baseline [6] results, and the last four columns report our results.
>
> |  No.  | Baseline [6]-ADE$_1$ | FDE$_1$ | ADE$_5$ | FDE$_5$ | Ours-ADE$_1$ | FDE$_1$ | ADE$_5$ | FDE$_5$ |
> |:-----:|:---------------:|:-------:|:-------:|:-------:|:------------:|:-------:|:-------:|:-------:|
> |   1   | 3.7141          | 1.6779  | 8.7896  | 3.5351  | 2.6286       | 6.6382  | 1.1055  | 2.7691  |
> |   2   | 3.6947          | 1.6733  | 8.7193  | 3.4978  | 2.6908       | 6.2563  | 1.2898  | 2.7164  |
> |   3   | 3.7732          | 1.6842  | 8.9267  | 3.5500  | 2.6556       | 6.4614  | 1.1795  | 2.7256  |
> |   4   | 3.7670          | 1.6646  | 8.8636  | 3.4412  | 2.8260       | 6.4106  | 1.2446  | 2.7754  |
> |   5   | 3.6500          | 1.6701  | 8.5636  | 3.4879  | 2.6475       | 6.5800  | 1.2297  | 2.7115  |
>
> All four metrics yield $p$-values less than 0.01, indicating that our method significantly outperforms the baseline.
>
> (2) Meanwhile, the evaluation metrics presented in Table 6 are averaged over five random seeds for Argoverse and nuScenes and three seeds for Waymo.
>
> Table 6. Multi-seed results on Argoverse (A), nuScenes (N), and Waymo (W). Metrics are reported as mean ± standard deviation.
> |     Metric    |   ADE$_1$    |   FDE$_1$    |   ADE$_5$    |   FDE$_5$    |  ADE$_{10}$   |  FDE_{10}$   |
> |:-------------:|:---------:|:---------:|:---------:|:---------:|:--------:|:--------:|
> |       N       | 2.69±0.08 | 6.47±0.15 | 1.21±0.07 | 2.74±0.03 | 0.85±0.02 | 2.00±0.01 |
> |    Metric     |   ADE$_1$    |   FDE$_1$    |   ADE$_6$    |   FDE$_6$    | minADE   | minFDE   |
> |     A & W     | 1.17±0.05 | 2.85±0.11 | 0.62±0.02 | 0.95±0.01 | 0.5852±0.0152 | 1.1431±0.0227 |
>
> [6] Pei Xu, et al. Context-aware timewise vaes for real-time vehicle trajectory prediction. IEEE Robotics and Automation Letters, 2023.

---

> > ### Comment · Reviewer_Hx7U · 2025-08-02
> >
> > Thank you for providing detailed descriptions for each of the points I raised. I find the answers satisfying.

---

> > > ### Author Response · Authors · 2025-08-06
> > >
> > > Thank you very much for your valuable review and for taking the time to provide detailed feedback. We sincerely appreciate your constructive suggestions, which have helped improve the manuscript. If there remain any concerns or clarifications needed, we would be delighted to address them further.

---

### Official Review · Reviewer_ejCn · 2025-07-04

**Clarity:** 3
**Significance:** 2
**Originality:** 3
**Rating:** 4
**Confidence:** 5

**Summary:**

This paper introduces the DSA framework, a novel approach for vehicle trajectory prediction. The core contribution is the systematic integration of human driving styles—categorized as conservative, aggressive, and normal —into the prediction model. The framework leverages the principles of KANs and the weierstrass approximation theorem. It proposes matching specific polynomial basis functions to each driving style based on their mathematical properties: Bernstein polynomials for smooth, conservative trajectories; Chebyshev polynomials for abrupt, aggressive trajectories; and Legendre polynomials for moderately fluctuating, normal trajectories. The model uses a MoE structure to create a weighted combination of these style-specific predictors. Furthermore, it employs a degree adaptation mechanism to optimize the complexity of the polynomial functions. The authors demonstrate through experiments on the nuScenes, Argoverse, and Waymo datasets that their DSA framework achieves state-of-the-art performance on multiple standard evaluation metrics.

**Questions:**

See weaknesses.

**Ethical Concerns:**

["NO or VERY MINOR ethics concerns only"]

**Final Justification:**

After reading the rebuttal and comments from other reviewers, my final rating is borderline accept.

**Limitations:**

Yes

**Quality:**

2

**Strengths And Weaknesses:**

Strengths:
Propose a systematic and theoretically-grounded framework that explicitly matches different classes of polynomial basis functions to distinct, predefined driving styles for trajectory prediction. Extensive experiments on three major datasets demonstrate SOTA performance. Thorough ablation studies to validate their design choices. The experiments effectively demonstrate the benefit of combining all three driving style experts over using just one or two, and validate the specific choices of polynomials against other alternatives.
Weaknesses:
•	Lack of clarity regarding the concrete implementation of the core components.  Could you please elaborate on the precise architecture of the Expert networks? Specifically, how are the assigned polynomial basis functions (Bernstein, Chebyshev, Legendre) integrated?
•	The CAN taxonomy ignores continuous/style-blending behaviors (e.g., moderately aggressive) and relies on heuristic categorization. Real driving styles are context-dependent, but DSA uses fixed polynomial mappings. Evaluations focus on highway-like scenarios (nuScenes/Waymo). Performance in unstructured environments (e.g., pedestrians, intersections) is unverified.
•	Degree Adaptation is potentially misleading. The method describes using SMAC3 to solve an optimization problem for the degree n based on validation set performance. This sounds like an offline hyperparameter tuning step that finds a single optimal degree for each driving style, which is then fixed. If the degree is tuned offline, calling it "adaptation" is an overstatement. If it is adapted dynamically for each sample during inference, the mechanism and its computational overhead are not explained at all.

---

> ### Author Rebuttal · Authors · 2025-07-30
>
> We thank the reviewer ejCn for raising thought-provoking questions, which explanations and significantly improve the quality of the paper.
>
> Weaknesses & Question 1: The precise architecture of the Expert networks? How are the assigned polynomial basis functions integrated?
>
> A: The combined feature $z$ is integrated by computing the weighted sum of the expert outputs $E$, using the gating network $G$ as weights.
>
> In Polynomial Combination algorithm (Expert networks), each expert $E_j,(j=1,2,3)$ is implemented as a KAN-based module corresponding to a specific polynomial basis (Bernstein, Chebyshev and Legendre), each associated with a distinct driving style..
>
> For any input vehicle trajectory $X_i (i=1,\cdots,N)$, we first compute its score with respect to each expert using the following equation:
>
> $$
> H_{j}\leftarrow (X\cdot W_{g})_{i}+\mathsf{N} (0,1)\cdot \mathrm{Softplus} \left[ \left( X\cdot W_{n}\right)  \right]_{i},
> $$
>
> where $W_g$ and $W_n$ are trainable weight matrices (i.e., implemented as \texttt{nn.Linear} layers), and $\mathcal{N} (0,1)$ denotes standard Gaussian noise.
>
> Next, we compute the weight for each expert via the Softmax function:
>
>
> $G_{j}= \text{Softmax}(H)$, where $ H=\left( H_{1},H_{2},H_{3}\right)  $.
>
> Finally, the combined feature representation $z^{\text{Com}}_i$ for vehicle $i$ in the DSA Polynomial Combination block is computed as a weighted sum over the experts:
>
> $$
> z^{\text{Com} }_{i}=\sum^{3}_{j=1} G_{j}\left( X_{i}\right)  \cdot E_{j}\left( X_{i}\right) .
> $$
>
> 2: (1)The CAN taxonomy ignores continuous/style-blending behaviors (e.g., moderately aggressive) and relies on heuristic categorization.
>
> A: (1) We use three typical driving style categories to represent two behavioral extremes (Aggressive, Conservative) and an intermediate type (normal), following the convention established in prior works [1–4]. For more nuanced behaviors that fall between these categories (e.g., moderately aggressive), the weights of the three polynomial experts in the MoE-TopK network can be adaptively adjusted.
>
> [1] Yao X, Calvert S C, Hoogendoorn S P. Driving heterogeneity identification using machine learning: A review and framework for analysis[J]. Transportation Research Interdisciplinary Perspectives, 2025.
>
> [2] Zheng X, Yang P, Duan D, et al. Real-time driving style classification based on short‐term observations[J]. IET Communications, 2022.
>
> [3] Garrosa M, Olmeda E, Fuentes del Toro S, et al. Holistic vehicle instrumentation for assessing driver driving styles[J]. Sensors, 2021.
>
> [4] Xu L, Hu J, Jiang H, et al. Establishing style-oriented driver models by imitating human driving behaviors[J]. IEEE Transactions on Intelligent Transportation Systems, 2015.
>
>
> 2: (2) Real driving styles are context-dependent, but DSA uses fixed polynomial mappings.
>
> A: (2) We justify the choice of "fixed polynomial" types using both mathematical theory (**theoretical**) and empirical **experiment** results.
>
> ## Theoretical Justification
>
> We establish a theoretical matching between the three driving styles and corresponding polynomial bases based on the behavioral characteristics of each style:
>
> -**Conservative drivers** tend to maintain the lowest average speed and exhibit minimal behavioral changes. Their trajectories are smooth and stable with few abrupt accelerations or decelerations. The uniform convergence property of Bernstein polynomials ($B_n$) ensures that the approximation error decreases consistently across the interval, making them well-suited for approximating conservative trajectories.
>
> -**Aggressive drivers**, in contrast, display higher speeds and more abrupt behavioral changes, resulting in trajectories that are less smooth and more prone to sudden shifts. Chebyshev polynomials ($T_n$) which minimize the maximum approximation error, are particularly effective in handling sharp variations and limiting the influence of extreme points.
>
> -**Normal drivers** represent a balance between conservative and aggressive styles. Their trajectories tend to maintain moderate speeds and accelerations with regular but smooth behavioral changes. Legendre polynomials ($L_n$) known for their orthogonality and optimal approximation under the $L^2$ norm, are computationally efficient and capable of capturing continuous and moderately fluctuating trajectory patterns.
>
> ## Experiment Results
> We replace each original polynomial in ours with alternative types, including Charlier ($C_n$), Hermite ($H_n$) and Second-order ($S_n$) polynomials. The best-performing results (in bold) are highlighted in Table1.
>
> Table 1. Evaluate different p_n in the DSA framework on the Argoverse dataset. The "Replace" column indicates the original → replacement mapping and the corresponding driving style.
>
> | Method                  | Replace        | ADE$_1$    |   FDE$_1$    |   ADE$_6$    |   FDE$_6$   |
> |-------------------------|----------------|------|------|------|------|
> | $C_n+T_n+L_n$            | $B_n\rightarrow C_n$ Con    | 1.48 | 3.76 | 1.10 | 1.53 |
> | $B_n+S_n+L_n$            | $T_n\rightarrow S_n$ Agg    | 1.59 | 3.91 | 0.86 | 1.41 |
> | $B_n+T_n+H_n$            | $L_n\rightarrow H_n$ Nor    | 1.62 | **2.81** | 0.71 | 1.02 |
> | **DSA**                 | -              | **1.17** | 2.85 | **0.62** | **0.96** |
>
> From Table 1, our DSA framework yields the largest improvements: 27.8%, 43.6% and 37.3% in key metrics when using the original polynomial assignments. Although the combination $B_n+T_n+H_n$ results in a slightly lower FDE$_1$ (from 2.81 to 2.85, a 1.4% difference), the original DSA configuration still ranks second for FDE$_1$, confirming its overall robustness and effectiveness.
>
> 3: Evaluations focus on (1) highway-like scenarios (nuScenes/Waymo). Performance in unstructured environments (e.g., (2) pedestrians, intersections) is unverified.
>
> A: (1) We evaluate our framework on three real-world datasets that span both urban and suburban environments: (i) Boston and Singapore (nuScenes [5]), (ii) Miami and Pittsburgh (Argoverse [6]) and (iii) Phoenix, AZ, Kirkland, and other cities (Waymo [7]). The maps provided in these datasets include diverse road geometries, including standard 3-way and 4-way intersections, as well as T-intersections.
>
> [5] https://www.nuscenes.org/nuscenes.
>
> [6] https://www.argoverse.org/av1.html#overview-link.
>
> [7] https://waymo.com/blog/2019/08/waymo-open-dataset-sharing-our-self.
>
> (2) Our current work focuses on vehicle trajectory prediction. Thank you for your valuable suggestion, we plan to extend our framework to predict the trajectories of other traffic participants, such as pedestrians and cyclists in future work.
>
> 4: Degree Adaptation is potentially misleading. The method describes using SMAC3 to solve an optimization problem for the degree $n$ based on validation set performance. This sounds like an offline hyperparameter tuning step that finds a single optimal degree for each driving style, which is then fixed. If the degree is tuned offline, calling it "adaptation" is an overstatement. If it is adapted dynamically for each sample during inference, the mechanism and its computational overhead are not explained at all.
>
> A: We will revise Section 3.3.2 to be titled "Degree Adaptation". During the training process, our framework learns the optimal polynomial degree for each expert (corresponding to a driving style). During inference, the learned degree is used to determine suitable expert weights, analogous to how attention weights are computed in Transformer architectures.

---

> > ### Comment · Reviewer_ejCn · 2025-08-06
> > **response to rebuttal**
> >
> > Thanks for the detailed rebuttal, it partially addresses my concerns.

---

> > > ### Author Response · Authors · 2025-08-06
> > >
> > > We sincerely appreciate your thoughtful and professional
> > > feedback, your points are highly constructive. If any aspect remains unclear, we would be glad to offer further clarification.

---

### Note · Authors · 2025-08-15

We sincerely appreciate all reviewers for their recognition, constructive suggestions, and insightful inspiration regarding our framework. The discussion is of profound significance not only for this submission but also for our broader research.

We propose a Driving-Style-Adaptive (DSA) framework that integrates heterogeneous driving behaviors into trajectory prediction by matching driving styles to basis functions $p_n$ and adaptively combining them. DSA offers interpretable insights and experiments on three public datasets show that it improves prediction accuracy.

We appreciate all reviewers' comments highlighting the strengths of our work. In summary:

Reviewer 3sDS notes that incorporating driving behavior into deep learning is a solid approach, with consistent reasoning throughout the text. The results are presented in well-organized tables and figures and the paper's writing is clear and easy to understand.

Reviewer ejCn states that our work proposes a systematic and theoretically grounded framework that explicitly matches different classes of polynomial basis functions $p_n$. The experiments effectively demonstrate the benefits of combining specific choices of $p_n$.

Reviewer Hx7U notes that the proposed method is both interesting and novel. The paper is well written, with good flow and clarity. The introduction is well structured and the experimental section is explained clearly.

Reviewer 5qot highlights that the use of distinct $p_n$ bases is conceptually linked to driving behavior characteristics. The dynamic combination of styles enhances the model's ability to handle diverse, multimodal behaviors, and the experiments are extensive.

The core concerns and our corresponding solutions (detailed by reviewer name and number) are as follows:

1.	Relationship between driving styles and specific $p_n$:
We present a mathematical analysis [5qot W3; ejCn 2(2)] and experimental results evaluating the trajectory cosine similarity between the highest-weighted $p_n$ and the predefined driving style standards [5qot Q3 & W2; 3sDS 1(1)].

2. How expert weights reflect these behaviors:
We compute the correct matching rate, defined as the percentage of cases in which the corresponding expert has the highest weight under each driving style [Hx7U W2(2); 3sDS 1(2)].

We sincerely thank the reviewers for their invaluable insights and will address all suggested revisions meticulously in the revised manuscript.

---

### Decision · Program_Chairs · 2025-09-17

**Decision:**

Accept (poster)

**Comment:**

This paper proposes the Driving-Style-Adaptive (DSA) framework for vehicle trajectory prediction, addressing the limitation that existing methods fail to account for trajectory heterogeneity from different driving styles. The core claim is that trajectories can be more accurately predicted by modeling driving behaviors through style-specific polynomial basis functions within a Kolmogorov-Arnold Network (KAN) architecture. The framework categorizes driving into three types: conservative (Bernstein polynomials), aggressive (Chebyshev polynomials), and normal (Legendre polynomials), combined through a mixture-of-experts mechanism. Experimental results show state-of-the-art performance across three datasets with 80-94% correct style-to-expert matching rates and statistically significant improvements over baselines.

The main strengths are as follows: (1) Novelty: the systematic integration of driving style modeling into trajectory prediction represents a conceptually sound advancement. The theoretical matching between polynomial properties and driving characteristics provides mathematical justification for the approach. (2) Comprehensive experimental validation: the evaluation spans three major real-world datasets with diverse geographic and traffic conditions. Multi-seed experiments with error bars demonstrate result stability, and statistical significance testing confirms improvements over baselines. (3) Interpretability: unlike black-box approaches, the framework provides explainable insights through expert weight analysis, showing which driving styles contribute to specific predictions. (4) Thorough ablation studies: The paper includes extensive ablation experiments validating the choice of polynomial types, the three-expert configuration, and the degree adaptation mechanism. (5) Strong empirical results: Consistent improvements across multiple metrics with appropriate statistical validation demonstrate the framework's effectiveness.

The weaknesses are as follows:  (1) Lack of clarity regarding the concrete implementation of the core components. (2) Some descriptions are misleading. For example, "degree adaptation" is potentially misleading. (3) The paper lacks a visual analysis, and affects the model's interpretability. The contribution of each polynomial expert is not visualized or analyzed across scenarios, making it unclear if driving styles are explicitly learned or used during inference. (4) The paper does not appear to discuss training time or provide key implementation details.  (5) Lack of tests to assess whether the performance improvements are statistically meaningful. Including such tests would strengthen the empirical claims and support the reported gains more robustly. (6) Although the evaluation is comprehensive, it lacks error bars on the presented results.

The reasons for acceptance: (1) The paper addresses a genuine limitation in existing trajectory prediction methods by incorporating driving style heterogeneity. The approach is technically sound and demonstrates measurable improvements. (2)  The comprehensive evaluation methodology, including multi-seed experiments, statistical significance testing, and extensive ablation studies, provides strong evidence for the claims. (3) The interpretability aspect adds significant value for safety-critical autonomous driving applications where understanding predictions rationale is crucial. (4) The main concerns of reviewers have been well addressed in the rebuttal period.

The discussion during rebuttal: (1) Reviewer ejCn raised concerns about implementation details, categorical driving styles, and misleading "degree adaptation" terminology. The authors provided detailed mathematical formulations and theoretical justification for polynomial choices, though fundamental limitations in categorical modeling and terminology issues remain. (2)  Reviewer Hx7U identified issues with typos, missing visual analysis, and lack of significance testing. The authors addressed these with multi-seed results, statistical analysis, and detailed implementation specifications, leading this reviewer to a clear acceptance.  (3) Reviewer 5qot questioned multi-modal prediction mechanisms and interpretability analysis. The authors clarified parallel generation processes, provided expert selection analysis, and included clustering validation supporting their design choices. (4) Reviewer 3sDS requested error bars, limitations discussion, and broader contextual considerations. The authors provided comprehensive multi-seed results and outlined extensive future work directions. The rebuttal period was productive, with authors providing substantial additional experimental results including clustering analysis and significance testing.